# On the objectivity, reliability, and validity of deep learning enabled bioimage analyses

Dennis Segebarth[1†], Matthias Griebel[2†], Nikolai Stein[2], Cora R von Collenberg[1], Corinna Martin[1], Dominik Fiedler[3], Lucas B Comeras[4], Anupam Sah[5], Victoria Schoeffler[6], Teresa Lüffe[6], Alexander Dürr[2], Rohini Gupta[1], Manju Sasi[1], Christina Lillesaar[6], Maren D Lange[3], Ramon O Tasan[4], Nicolas Singewald[5], Hans-Christian Pape[3], Christoph M Flath[2‡*], Robert Blum[1,7‡*]

[1]Institute of Clinical Neurobiology, University Hospital Würzburg, Würzburg, Germany; [2]Department of Business and Economics, University of Würzburg, Würzburg, Germany; [3]Institute of Physiology I, Westfälische Wilhlems-Universität, Münster, Germany; [4]Department of Pharmacology, Medical University of Innsbruck, Innsbruck, Austria; [5]Department of Pharmacology and Toxicology, Institute of Pharmacy and Center for Molecular Biosciences Innsbruck, University of Innsbruck, Innsbruck, Austria; [6]Department of Child and Adolescent Psychiatry, Center of Mental Health, University Hospital Würzburg, Würzburg, Germany; [7]Comprehensive Anxiety Center, Würzburg, Germany

*For correspondence:
christoph.flath@uni-wuerzburg.de (CMF);
Blum_R@ukw.de (RB)

[†]These authors contributed equally to this work
[‡]These authors also contributed equally to this work

Competing interests: The authors declare that no competing interests exist.

**Abstract** Bioimage analysis of fluorescent labels is widely used in the life sciences. Recent advances in deep learning (DL) allow automating time-consuming manual image analysis processes based on annotated training data. However, manual annotation of fluorescent features with a low signal-to-noise ratio is somewhat subjective. Training DL models on subjective annotations may be instable or yield biased models. In turn, these models may be unable to reliably detect biological effects. An analysis pipeline integrating data annotation, ground truth estimation, and model training can mitigate this risk. To evaluate this integrated process, we compared different DL-based analysis approaches. With data from two model organisms (mice, zebrafish) and five laboratories, we show that ground truth estimation from multiple human annotators helps to establish objectivity in fluorescent feature annotations. Furthermore, ensembles of multiple models trained on the estimated ground truth establish reliability and validity. Our research provides guidelines for reproducible DL-based bioimage analyses.

## Introduction

Modern microscopy methods enable researchers to capture images that describe cellular and molecular features in biological samples at an unprecedented scale. One of the most frequently used imaging methods is fluorescent labeling of biological macromolecules, both in vitro and in vivo. In order to test a biological hypothesis, fluorescent features have to be interpreted and analyzed quantitatively, a process known as bioimage analysis (*Meijering et al., 2016*). However, fluorescence does not provide clear signal-to-noise borders, forcing human experts to utilize individual heuristic criteria, such as morphology, size, or signal intensity to classify fluorescent signals as background, or to, often manually, annotate them as a region of interest (ROI). This cognitive decision process depends on the graphical perception capabilities of the individual annotator (*Cleveland and McGill, 1985*). Constant technological advances in fluorescence microscopy facilitate the automatized

**eLife digest** Research in biology generates many image datasets, mostly from microscopy. These images have to be analyzed, and much of this analysis relies on a human expert looking at the images and manually annotating features. Image datasets are often large, and human annotation can be subjective, so automating image analysis is highly desirable. This is where machine learning algorithms, such as deep learning, have proven to be useful. In order for deep learning algorithms to work first they have to be 'trained'. Deep learning algorithms are trained by being given a training dataset that has been annotated by human experts. The algorithms extract the relevant features to look out for from this training dataset and can then look for these features in other image data.

However, it is also worth noting that because these models try to mimic the annotation behavior presented to them during training as well as possible, they can sometimes also mimic an expert's subjectivity when annotating data. Segebarth, Griebel et al. asked whether this was the case, whether it had an impact on the outcome of the image data analysis, and whether it was possible to avoid this problem when using deep learning for imaging dataset analysis.

For this research, Segebarth, Griebel et al. used microscopy images of mouse brain sections, where a protein called cFOS had been labeled with a fluorescent tag. This protein typically controls the rate at which DNA information is copied into RNA, leading to the production of proteins. Its activity can be influenced experimentally by testing the behaviors of mice. Thus, this experimental manipulation can be used to evaluate the results of deep learning-based image analyses.

First, the fluorescent images were interpreted manually by a group of human experts. Then, their results were used to train a large variety of deep learning models. Models were trained either on the results of an individual expert or on the results pooled from all experts to come up with a consensus model, a deep learning model that learned from the personal annotation preferences of all experts. This made it possible to test whether training a model on multiple experts reduces the risk of subjectivity. As the training of deep learning models is random, Segebarth, Griebel et al. also tested whether combining the predictions from multiple models in a so-called model ensemble improves the consistency of the analyses. For evaluation, the annotations of the deep learning models were compared to those of the human experts, to ensure that the results were not influenced by the subjective behavior of one person. The results of all bioimage annotations were finally compared to the experimental results from analyzing the mice's behaviors in order to check whether the models were able to find the behavioral effect on cFOS.

Segebarth, Griebel et al. concluded that combining the expert knowledge of multiple experts reduces the subjectivity of bioimage annotation by deep learning algorithms. Combining such consensus information in a group of deep learning models improves the quality of bioimage analysis, so that the results are reliable, transparent and less subjective.

acquisition of large image datasets, even at high resolution and with high throughput (*Li et al., 2010*; *McDole et al., 2018*; *Osten and Margrie, 2013*). The ever increasing workload associated with image feature annotation therefore calls for computer-aided automated bioimage analysis. However, attempts to replace human experts and to automate the annotation process using traditional image thresholding techniques (e.g. histogram shape-, entropy-, or clustering-based methods [*Sezgin and Sankur, 2004*]) frequently lack flexibility, as they rely on a high signal-to-noise ratio in the images or require computational expertise for user-based adaptation to individual datasets (*von Chamier et al., 2019*). In recent years, deep learning (DL) and in particular deep convolutional neural networks have shown remarkable capacities in image recognition tasks, opening new possibilities to perform automatized image analysis. DL-based approaches have emerged as an alternative to conventional feature annotation or segmentation methods (*Caicedo et al., 2019*) and are even capable of performing complex tasks such as artificial labeling of plain bright-field images (*von Chamier et al., 2019*; *Christiansen et al., 2018*; *Ounkomol et al., 2018*). The main difference between conventional and DL algorithms is that conventional algorithms follow predefined rules (hard-coded), while DL algorithms are flexible to learn the respective task on base of a training dataset (*LeCun et al., 2015*). Yet, deployment of DL approaches necessitates both computational

expertise and suitable computing resources. These requirements frequently prevent non-AI experts from applying DL to routine image analysis tasks. Initial efforts have already been made to break down these barriers, for instance, by integration into prevalent bioimaging tools such as *ImageJ* (*Falk et al., 2019*) and *CellProfiler* (*McQuin et al., 2018*), or using cloud-based approaches (*Haberl et al., 2018*). To harness the potentials of these DL-based methods, they require integration into the bioimage analysis pipeline. We argue that such an integration into the scientific process ultimately necessitates DL-based approaches to meet the same standards as any method in an empirical study. We can derive these standards from the general quality criteria of qualitative and quantitative research: objectivity, reliability, and validity (*Frambach et al., 2013*).

*Objectivity* refers to the neutrality of evidence, with the aim to reduce personal preferences, emotions, or simply limitations introduced by the context in which manual feature annotation is performed (*Frambach et al., 2013*). Manual annotation of fluorescent features has long been known to be subjective, especially in the case of weak signal-to-noise thresholds (*Schmitz et al., 1999*; *Collier et al., 2003*; *Niedworok et al., 2016*). Notably, there is no objective ground truth reference in the particular case of fluorescent label segmentation, causing a critical problem for training and evaluation of DL algorithms. As multiple studies have pointed out that annotations of low quality can cause DL algorithms to either fail to train or to reproduce inconsistent annotations on new data (*von Chamier et al., 2019*; *Falk et al., 2019*), this is a crucial obstacle for applying DL to bioimage analysis processes.

*Reliability* is concerned with the consistency of evidence (*Frambach et al., 2013*). To allow an unambiguous understanding of this concept, we further distinguish between repeatability and reproducibility. Repeatability or test-retest reliability is defined as 'closeness of the agreement between the results of successive measurements of the same measure and carried out under the same conditions' (*Taylor and Kuyatt, 1994*, 14), which is guaranteed for any deterministic DL model. Reproducibility, on the other hand, is specified as 'closeness of the agreement between the results of measurements of the same measure and carried out under changed conditions' (*Taylor and Kuyatt, 1994*, 14), for example, different observer, or different apparatus. This is a critical point, since the output of different DL models trained on the same training dataset can vary significantly. This is caused by the stochastic training procedure (e.g. random initialization, random sampling and data augmentation [*Ronneberger et al., 2015*]), the choice of model parameters (e.g. model architecture, weights, activation functions), and the choice of hyperparameters (e.g. learning rate, mini-batch size, training epochs). Consequently, the reproducibility of DL models merits careful investigation.

Finally, *validity* relates to the truth value of evidence, that is, whether we in fact measured what we intended to. Moreover, validity implies reliability - but not vice versa (*Frambach et al., 2013*). On a basis of a given ground truth, validity is typically measured using appropriate similarity measures such as F1 score for detection and Intersection over Union (IoU) for segmentation purposes (*Ronneberger et al., 2015*; *Falk et al., 2019*; *Caicedo et al., 2019*). In addition, the DL community has established widely accepted standards for training models. These comprise, among other things, techniques to avoid overfitting (regularization techniques and cross-validation), tuning hyperparameters, and selecting appropriate metrics for model evaluation. However, these standards do not apply for the training and evaluation of a DL model in the absence of a ground truth, like in the case of fluorescent features.

Taken together and with regard to the discussion about a reproducibility crisis in the fields of biology, medicine and artificial intelligence (*Siebert et al., 2015*; *Baker, 2016*; *Ioannidis, 2016*; *Hutson, 2018*; *Fanelli, 2018*; *Chen et al., 2019*), these limitations indicate that DL could aggravate this crisis by adding even more unknowns and uncertainties to bioimage analyses.

However, the present study asks whether DL, if instantiated in an appropriate manner, also holds the potential to instead enhance the objectivity, reproducibility and validity of bioimage analysis. To tackle this conundrum, we investigated different DL-based strategies on five fluorescence image datasets. We show that training of DL models on the pooled input of multiple human experts utilizing ground truth estimation (consensus models) increases objectivity of fluorescent feature segmentation. Furthermore, we demonstrate that ensembles of consensus models are even capable of enhancing the reliability and validity of bioimage analysis of ambiguous image data, such as fluorescence features in histological tissue sections.

# Results

In order to evaluate the impact of DL on bioimage analysis results, we instantiated three exemplary DL-based strategies (*Figure 1*; strategies color-coded in gray, blue, and orange) and investigate them in terms of objectivity, reliability, and validity of fluorescent feature annotation. The first strategy, *expert models* (gray), reflects mere automation of the annotation process of fluorescent features in microscopy images. Here, manual annotations of a single human expert are used to train an individual (and hence expert-specific) DL model with a U-Net (*Ronneberger et al., 2015*) architecture. U-Net and its variants have emerged as the de facto standard for biomedical image segmentation purposes (*McQuin et al., 2018*; *Falk et al., 2019*; *Caicedo et al., 2019*). The second strategy, consensus *models* (blue), addresses the objectivity of signal annotations. Contrary to the first strategy, simultaneous truth and performance level estimation (STAPLE) (*Warfield et al., 2004*) is used to estimate a ground truth and create consensus annotations. The estimated ground truth (est. GT) annotation reflects the pooled input of multiple human experts and is therefore thought to be less affected by a potential subjective bias of a single expert. We then train a single U-Net model to create a consensus model. The third strategy, consensus *ensembles* (orange), seeks to ensure reliability and eventually validity. Going beyond the second strategy, we train multiple consensus U-Net models to create a consensus ensemble. Such model ensembles are known to be more robust to noise (*Dietterich, 2000*). Hence, we hypothesize that the consensus ensembles mitigate the randomness

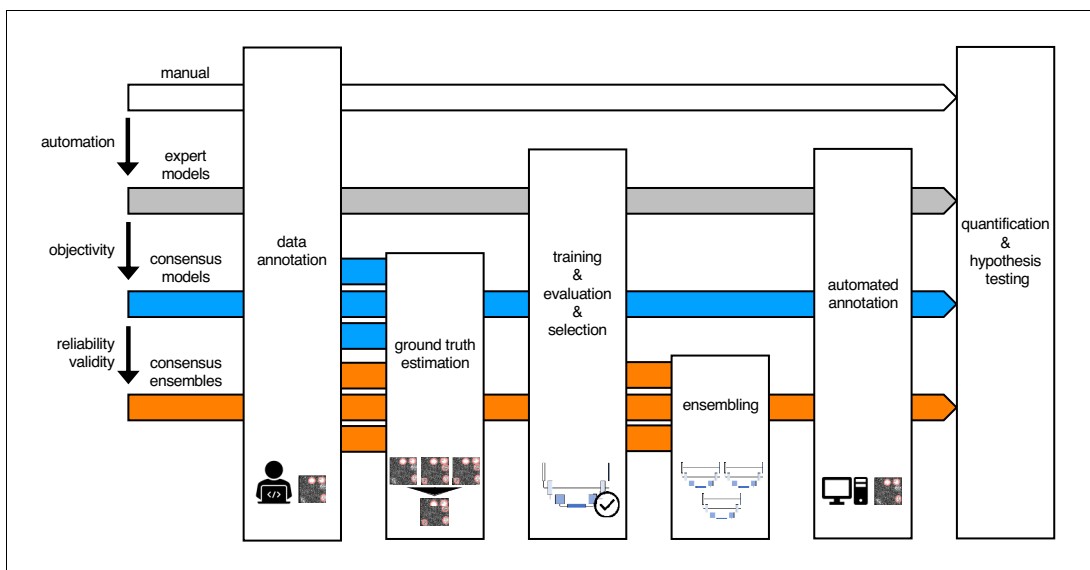

**Figure 1.** Schematic illustration of bioimage analysis strategies and corresponding hypotheses. Four bioimage analysis strategies are depicted. Manual (white) refers to manual, heuristic fluorescent feature annotation by a human expert. The three DL-based strategies for automatized fluorescent feature annotation are based on expert models (gray), consensus models (blue) and consensus ensembles (orange). For all DL-based strategies, a representative subset of microscopy images is annotated by human experts. Here, we depict labels of cFOS-positive nuclei and the corresponding annotations (pink). These annotations are used in either individual training datasets (gray: expert models) or pooled in a single training dataset by means of ground truth estimation from the expert annotations (blue: consensus models, orange: consensus ensembles). Next, deep learning models are trained on the training dataset and evaluated on a holdout validation dataset. Subsequently, the predictions of individual models (gray and blue) or model ensembles (orange) are used to compute binary segmentation masks for the entire bioimage dataset. Based on these fluorescent feature segmentations, quantification and statistical analyses are performed. The expert model strategy enables the automation of a manual analysis. To mitigate the bias from subjective feature annotations in the expert model strategy, we introduce the consensus model strategy. Finally, the consensus ensembles alleviate the random effects in the training procedure and seek to ensure reliability and eventually, validity.

The online version of this article includes the following figure supplement(s) for figure 1:

**Figure supplement 1.** U-Net architecture.
**Figure supplement 2.** Illustration of bioimage dataset Lab-Wue1.

in the training process. Moreover, deep ensembles are supposed to yield high-quality predictive uncertainty estimates (*Lakshminarayanan et al., 2017*).

For each of the three strategies, we complete the bioimage analysis by performing quantification and hypothesis testing on a typical fluorescent microscopy image dataset (*Figure 1—figure supplement 2*). These images describe changes in fluorescence signal abundance of a protein called cFOS in brain sections of mice. cFOS is an activity-dependent transcription factor and its expression in the brain can be modified experimentally by behavioral testing of the animals (*Gallo et al., 2018*). The low signal-to-noise ratio of this label, its broad usage in neurobiology and the well-established correlation of its abundance with behavioral paradigms render it an ideal bioimage dataset to test our hypotheses (*Shuvaev et al., 2017*; *Gallo et al., 2018*).

## Consensus ensembles yield the best results for validity and reproducibility metrics

The primary goal in bioimage analysis is to rigorously test a biological hypothesis. To leverage the potentials of DL models within this procedure, we need to trust our model – by establishing objectivity, reliability, and validity. Pertaining to the case of fluorescent labels, validity (measuring what is intended to be measured) requires objectivity to know what exactly we intend to measure in the absence of a ground truth. Similarly, reliability in terms of repeatability and reproducibility is a prerequisite for a valid and trustworthy model. Starting from the expert model strategy, we seek to establish objectivity (consensus models) and, successively, reliability and validity in the consensus ensemble strategy. In the following analysis, we first turn toward a comprehensive evaluation of the objectivity and its relation to validity before moving on to the concept of reliability.

To assess the three different strategies, a training dataset of 36 images and a test set of nine microscopy images (1024 × 1024 px, 1.61 px/μm, on average ~35 nuclei per image, see also *Figure 1—figure supplement 2*) showing cFOS immunoreactivity were manually annotated by five independent experts (experts 1–5). In absence of a rigorously objective ground truth, we used STAPLE (*Warfield et al., 2004*) to compute an estimated ground truth (est. GT) based on all expert annotations for each image. First, we trained a set of DL models on the 36 training images and corresponding annotations, either made by an individual human expert or as reflected in the est. GT (see Materials and methods for the data set and detailed training, evaluation and model selection strategy). Then, we used our test set to evaluate the segmentation (Mean IoU) and detection (F1 score) performance of human experts and all trained models by means of similarity analysis on the level of individual images.

For the pairwise comparison of annotations (segmentation masks), we calculated the intersection over union (IoU) for all overlapping pairs of ROIs between two segmentation masks (*Figure 2A*; see 7.9.1 Segmentation and detection). Following *Maška et al., 2014*, we consider two ROIs with an IoU of at least 0.5 as matching and calculated the F1 score $M_{F1score}$ as the harmonic mean of precision and recall (*Figure 2B*; see 7.9.1 Segmentation and detection). As bioimaging studies predominantly use measures related to counting ROIs in their analyses, we also focused on the feature detection performance ($M_{F1score}$). The color coding (gray, blue, orange) introduced in *Figure 2C* refers to the different strategies depicted in *Figure 1* and applies to all figures, if not indicated otherwise.

To better grasp the difficulties in annotating cFOS-positive nuclei as fluorescent features in these images, we first compared manual expert annotations (*Figure 2D*). The analysis revealed substantial differences between the annotations of the different experts and shows varying inter-rater agreement (*Schmitz et al., 1999*; *Collier et al., 2003*; *Niedworok et al., 2016*). The level of inter-rater variability was inversely correlated with the relative signal intensities (*Figure 2—figure supplement 1*; *Niedworok et al., 2016*).

By comparing the annotations of the expert models (gray) to the annotations of the respective expert (*Figure 2E*), we observed a higher $M_{F1score}$ median compared to the inter-rater agreement (*Figure 2D*) in the majority of cases. Furthermore, comparing the similarity analysis results of human experts with those of their respective expert-specific models revealed that they are closely related (*Figure 2F*, *Figure 2—figure supplement 3*, and *Figure 2—figure supplement 4*). As pointed out by *von Chamier et al., 2019*, this indicates that our expert models are able to learn and reproduce the annotation behavior of the individual experts. This becomes particularly evident in the annotations of the DL models trained on expert 1 (*Figure 2F*, *Figure 2—figure supplement 3*, and *Figure 2—figure supplement 4*).

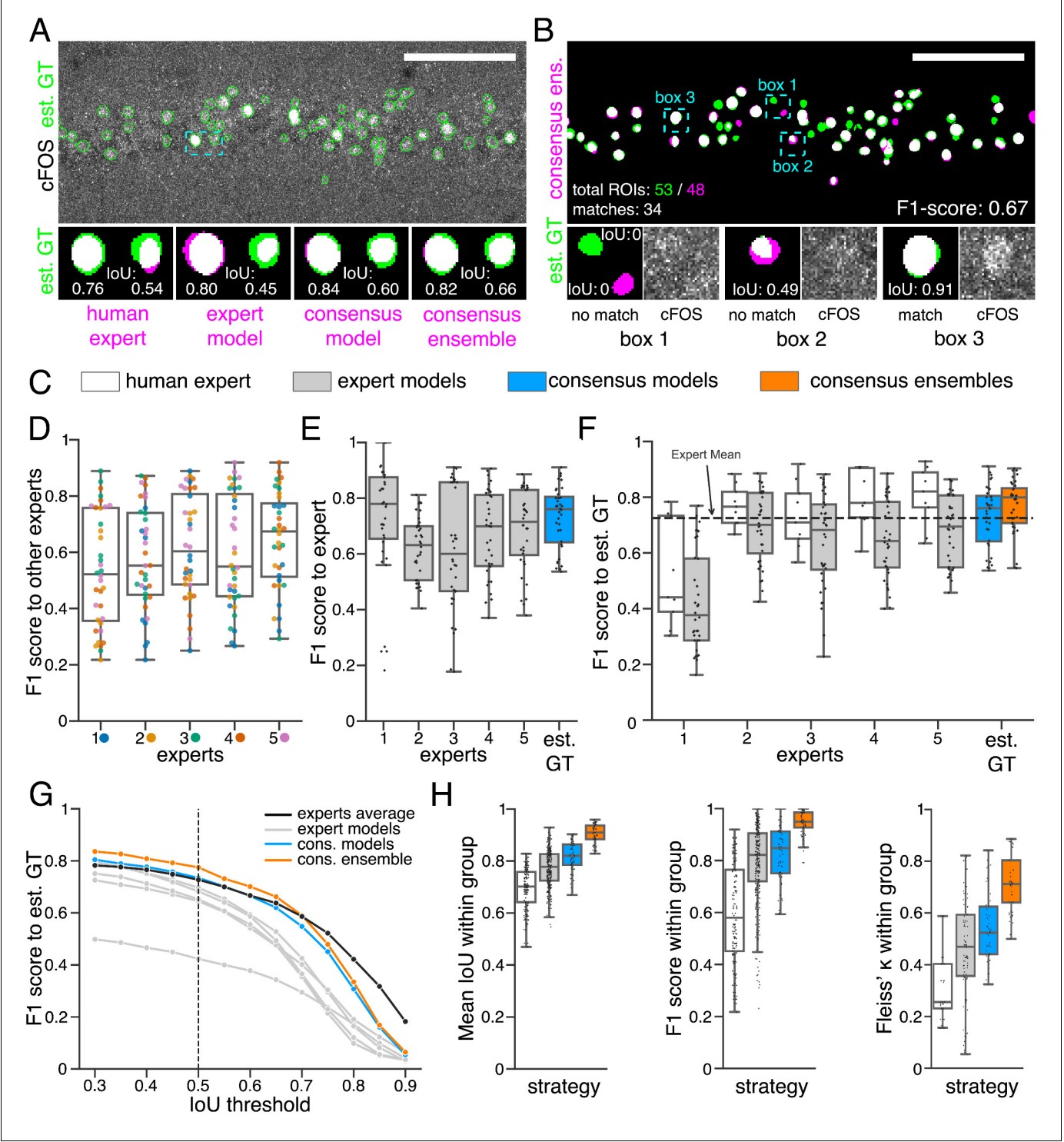

**Figure 2.** Similarity analysis of fluorescent feature annotations by manual or DL-based strategies. (**A**) Representative example of IoU $M_{IoU}$ calculations on a field of view (FOV) in a bioimage. Image raw data show the labeling of cFOS in a maximum intensity projection image of the CA1 region in the hippocampus (brightness and contrast enhanced). The similarity of estimated ground truth (est. GT) annotations (green), derived from the annotations of five expert neuroscientists, are compared to those of one human expert, an expert model, a consensus model, and a consensus ensemble (magenta, respectively). IoU results of two ROIs are shown in detail for each comparison (magnification of cyan box). Scale bar: 100 µm. (**B**) F1 score $M_{F1score}$ calculations on the same FOV as shown in (**A**). The est. GT annotations (green; 53 ROIs) are compared to those of a consensus ensemble (magenta; 48 ROIs). IoU-based matching of ROIs at an IoU-threshold of $t = 0.5$ is depicted in three magnified subregions of the image (cyan boxes 1-3). Scale bar: 100 µm. (**C–H**) All comparisons are performed exclusively on a separate image test set which was withheld from model training and validation. (**C**) Color

*Figure 2 continued on next page*

*Figure 2 continued*

coding refers to the individual strategies, as introduced in *Figure 1*: white: manual approach, gray: expert models, blue: consensus models, orange: consensus ensembles. (D) $M_{\text{F1score}}$ between individual manual expert annotations and their overall reliability of agreement given as the mean of Fleiss' $\kappa$. (E) $M_{\text{F1score}}$ between annotations predicted by individual models and the annotations of the respective expert (or est. GT), whose annotations were used for training. $N_{\text{models per expert}} = 4$. (F) $M_{\text{F1score}}$ between manual expert annotations, the respective expert models, consensus models, and consensus ensembles compared to the est. GT as reference. A horizontal line denotes human expert average. $N_{\text{models}} = 4$, $N_{\text{ensembles}} = 4$. (G) Means of $M_{\text{F1score}}$ of the individual DL-based strategies and of the human expert average compared to the est. GT plotted for different IoU matching thresholds $t$. A dashed line indicates the default threshold $t = 0.5$. $N_{\text{models}} = 4$, $N_{\text{ensembles}} = 4$. (H) Annotation reliability of the individual strategies assessed as the similarities between annotations within the respective strategy. We calculated $\bar{M}_{\text{IoU}}$, $M_{\text{F1score}}$ and Fleiss' $\kappa$. $N_{\text{experts}} = 5$, $N_{\text{models}} = 4$, $N_{\text{ensembles}} = 4$.

The online version of this article includes the following figure supplement(s) for figure 2:

**Figure supplement 1.** Extended subjectivity analysis.
**Figure supplement 2.** Ensemble size and reliability.
**Figure supplement 3.** Extended similarity analysis: F1 score.
**Figure supplement 4.** Extended similarity analysis: mean IoU.

Overall, the expert models yield a lower similarity to the est. GT compared to the consensus models (blue) or consensus ensembles (orange). Notably, both consensus models and consensus ensembles perform on par with human experts. Hereby, the consensus ensembles outperform all other strategies, even at varying IoU thresholds (*Figure 2F* and *Figure 2G*).

In order to test for reliability of our analysis, we measured the repeatability and reproducibility of fluorescent feature annotation of our DL strategies. We assumed that the repeatability is assured for all our strategies due to the deterministic nature of our DL models (unchanged conditions imply unchanged model weights). Hence, our evaluation was focused on the reproducibility, meaning the impact of the stochastic training process on the output. Inter-expert and inter-model comparisons within each strategy unveiled a better performance of the consensus ensembles strategy concerning both detection ($M_{\text{F1score}}$) and segmentation ($\bar{M}_{\text{IoU}}$) of the fluorescent features (*Figure 2H*). Calculating the Fleiss' kappa value (*Fleiss and Cohen, 1973*) revealed that consensus ensemble annotations show a high reliability of agreement (*Figure 2H*). Following the Fleiss' kappa interpretation from *Landis and Koch, 1977*, the results for the consensus ensembles indicate a substantial or almost perfect agreement. In contrast, the Fleiss' kappa values for human experts refer to a fair agreement while the results for the alternative DL strategies lead to a moderate agreement (*Figure 2H*).

In summary, the similarity analysis of the three strategies shows that training of DL models solely on the input of a single human expert imposes a high risk of incorporating an intrinsic bias and therefore resembles, as hypothesized, a mere automation of manual image annotation. Both consensus models and consensus ensembles perform on par with human experts regarding the similarity to the est. GT, but the consensus ensembles yield by far the best results regarding their reproducibility. We conclude that, in terms of similarity metrics, only the consensus ensemble strategy meet the bioimaging standards for objectivity, reliability, and validity.

## Consensus ensembles yield reliable bioimage analysis results

Similarity analysis is inevitable to assess the quality of a model's output, that is, the predicted segmentations (*Ronneberger et al., 2015*; *Caicedo et al., 2019*; *Falk et al., 2019*). However, the primary goal of bioimage analysis is the unbiased quantification of distinct image features that correlate with experimental conditions. So far, it has remained unclear whether objectivity, reliability, and validity for bioimage analysis can be inferred directly from similarity metrics.

In order to systematically address this question, we used our image dataset to quantify the abundance of cFOS in brain sections of mice after Pavlovian contextual fear conditioning. It is well established in the neuroscientific literature that rodents show changes in the distribution and abundance of cFOS in a specific brain region, namely the hippocampus, after processing information about places and contexts (*Keiser et al., 2017*; *Campeau et al., 1997*; *Huff et al., 2006*; *Ramamoorthi et al., 2011*; *Tayler et al., 2013*; *Murawski et al., 2012*; *Guzowski et al., 2001*). Consequently, our experimental dataset offered us a second line of evidence, the objective analysis of mouse behavior, in addition to the changes of fluorescent features to validate the bioimage analyses results of our DL-based strategies.

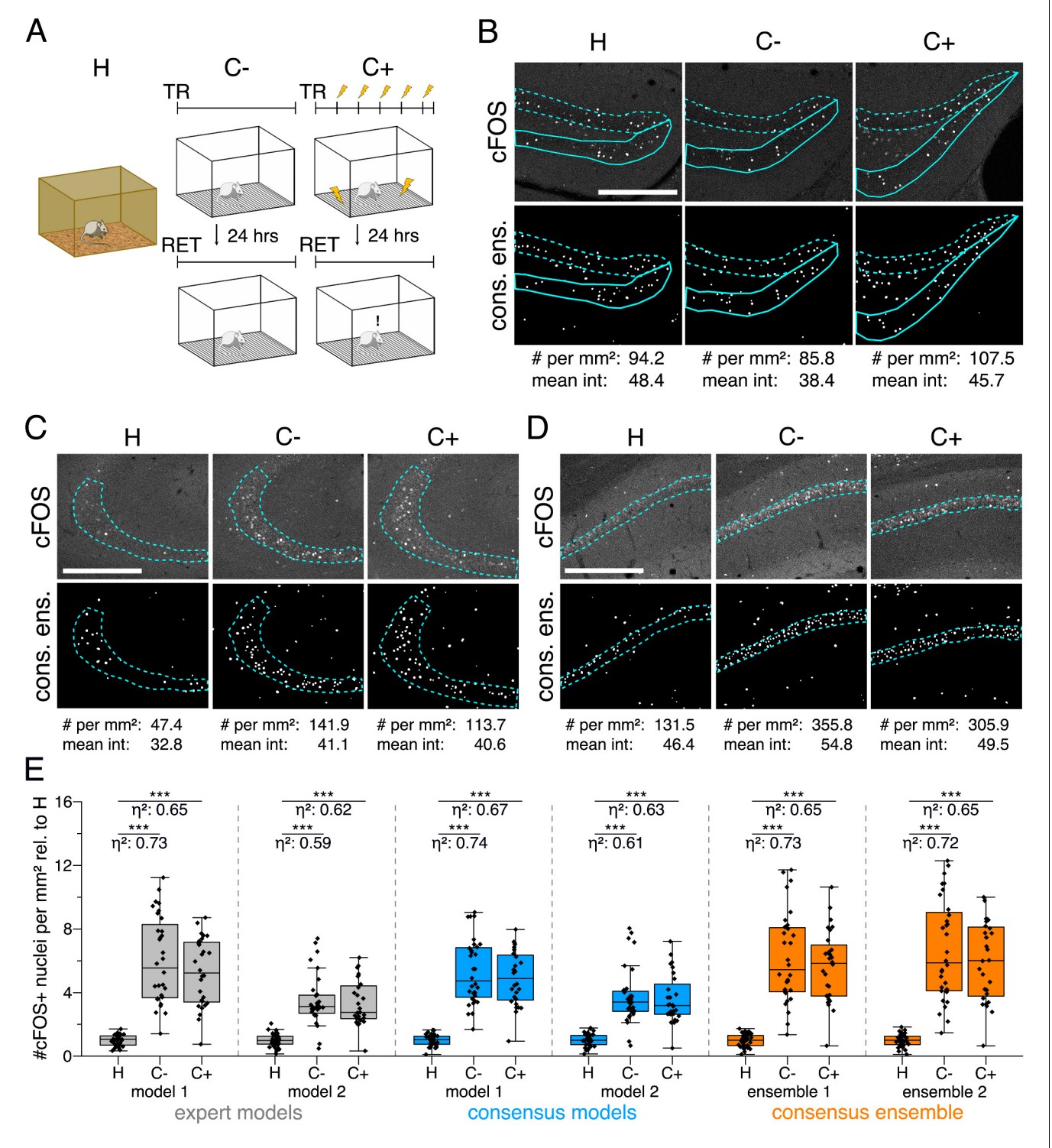

**Figure 3.** Application of different DL-based strategies for fluorescent feature annotation. The figure introduces how three DL-based strategies are applied for annotation of a representative fluorescent label, here cFOS, in a representative image data set. Raw image data show behavior-related changes in the abundance and distribution of the protein cFOS in the dorsal hippocampus, a brain center for encoding of context-dependent memory. (A) Three experimental groups were investigated: Mice kept in their homecage (H), mice that were trained to a context, but did not experience an electric foot shock (C-) and mice exposed to five foot shocks in the training context (C+). 24 hr after the initial training (TR), mice were re-exposed to the training context for memory retrieval (RET). Memory retrieval induces changes in cFOS levels. (B–D) Brightness and contrast enhanced maximum intensity projections showing cFOS fluorescent labels of the three experimental groups (H, C-, C+) with representative annotations of a consensus

*Figure 3 continued on next page*

*Figure 3 continued*

ensemble, for each hippocampal subregion. The annotations are used to quantify the number of cFOS-positive nuclei for each image (#) per mm$^2$ and their mean signal intensity (mean int., in bit-values) within the corresponding image region of interest, here the neuronal layers in the hippocampus (outlined in cyan). In B: granule cell layer (supra- and infrapyramidal blade), dotted line: suprapyramidal blade, solid line: infrapyramidal blade. In C: pyramidal cell layer of CA3; in D: pyramidal cell layer in CA1. Scale bars: 200 µm. (E) Analyses of cFOS-positive nuclei per mm$^2$, representatively shown for stratum pyramidale of CA1. Corresponding effect sizes are given as $\eta^2$ for each pairwise comparison. Two quantification results are shown for each strategy and were selected to represent the lowest (model 1 or ensemble 1) and highest (model 2 or ensemble 2) effect sizes (increase in cFOS) reported within each annotation strategy. Total analyses performed: $N_{expert\ models}$ = 20, $N_{consensus\ models}$ = 36, $N_{consensus\ ensembles}$ = 9. Number of analyzed mice (N) and images (n) per experimental condition: $N_H$ = 7, $N_{C-}$ = 7, $N_{C+}$ = 6; $n_H$ = 36, $n_{C-}$ = 32, $n_{C+}$ = 28. ***p<0.001 with Mann-Whitney-U test. Statistical data are available in *Figure 3—source data 1*.

The online version of this article includes the following source data and figure supplement(s) for figure 3:

**Source data 1.** Source files for analyses of cFOS-positive nuclei in CA1.
**Figure supplement 1.** Behavioral analysis *Lab-Wue1*.
**Figure supplement 1—source data 1.** Source files for behavioral analysis in *Figure 3—figure supplement 1*.

Our dataset comprised three experimental groups (*Figure 3A*). In one group, mice were directly taken from their homecage as naive learning controls (H). In the second group, mice were re-exposed to a previously explored training context as context controls (C-). Mice in the third group underwent Pavlovian fear conditioning and were also re-exposed to the training context (C+) (*Figure 3A*). These three groups of mice showed different behavioral responses. For instance, fear (threat; *LeDoux, 2014*) conditioned mice (C+) showed increased freezing behavior after fear acquisition and showed strong freezing responses when re-exposed to the training context 24 hr later (*Figure 3—figure supplement 1*). After behavioral testing, brain sections of the different groups of mice were prepared and labeled for the neuronal activity-related protein cFOS by indirect immunofluorescence. Sections were also labeled with the neuronal marker NeuN (Fox3), allowing the anatomical identification of hippocampal subregions of interest. Images were acquired as confocal microscopy image stacks (x,y-z) and maximum intensity projections were used for subsequent bioimage analysis (*Figure 1—figure supplement 2*). Overall, we quantified the number of cFOS-positive nuclei and their mean signal intensity in five regions of the dorsal hippocampus (DG as a whole, suprapyramidal DG, infrapyramidal DG, CA3, and CA1), and tested for significant differences between the three experimental groups (*Figure 3B–D*). To extend this analysis beyond hypothesis testing at a certain significance level, we calculated the effect size ($\eta^2$) for each of these 30 pairwise comparisons.

We illustrate our metrics with the detailed quantification of cFOS-positive nuclei in the stratum pyramidale of CA1 as a representative example and show two analyses for each DL strategy (*Figure 3E*). These two examples represent those two models of each strategy that yielded the lowest and the highest effect sizes, respectively (*Figure 3E*). Despite a general consensus of all models and ensembles on a context-dependent increase in the number of cFOS-positive nuclei, these quantifications already indicate that the variability of effect sizes decreases from expert models to consensus models and is lowest for consensus ensembles (*Figure 3E*).

The analysis in *Figure 4* allows us to further explore the impact of the different DL strategies on the bioimage analysis results for each hippocampal subregion. Here, we display a high-level comparison of the effect sizes and corresponding significance levels of 20 independently trained expert models (4 per expert), 36 consensus models, and 9 consensus ensembles (each derived from four consensus models). In contrast to the detailed illustration of selected models in *Figure 3E*, *Figure 4A*, for instance, summarizes the results for all analyses of the stratum pyramidale of CA1. As indicated before, all models and ensembles show a highly significant context-dependent increase in the number of cFOS-positive nuclei, but also a notable variation in effect sizes for both expert and consensus models. Moreover, we identify a significant context-dependent increase in the mean signal intensity of cFOS-positive nuclei for all consensus models and ensembles. The expert models, by contrast, yield a high variation in effect sizes at different significance levels. Interestingly, all four expert models trained on the annotations of expert 1 (and two other expert models only in the case of H vs. C+) did not yield a significant increase, indicating that expert 1's annotation behavior was incorporated into the expert-1-specific models and that this also affects the bioimage analysis results (*Figure 4A*).

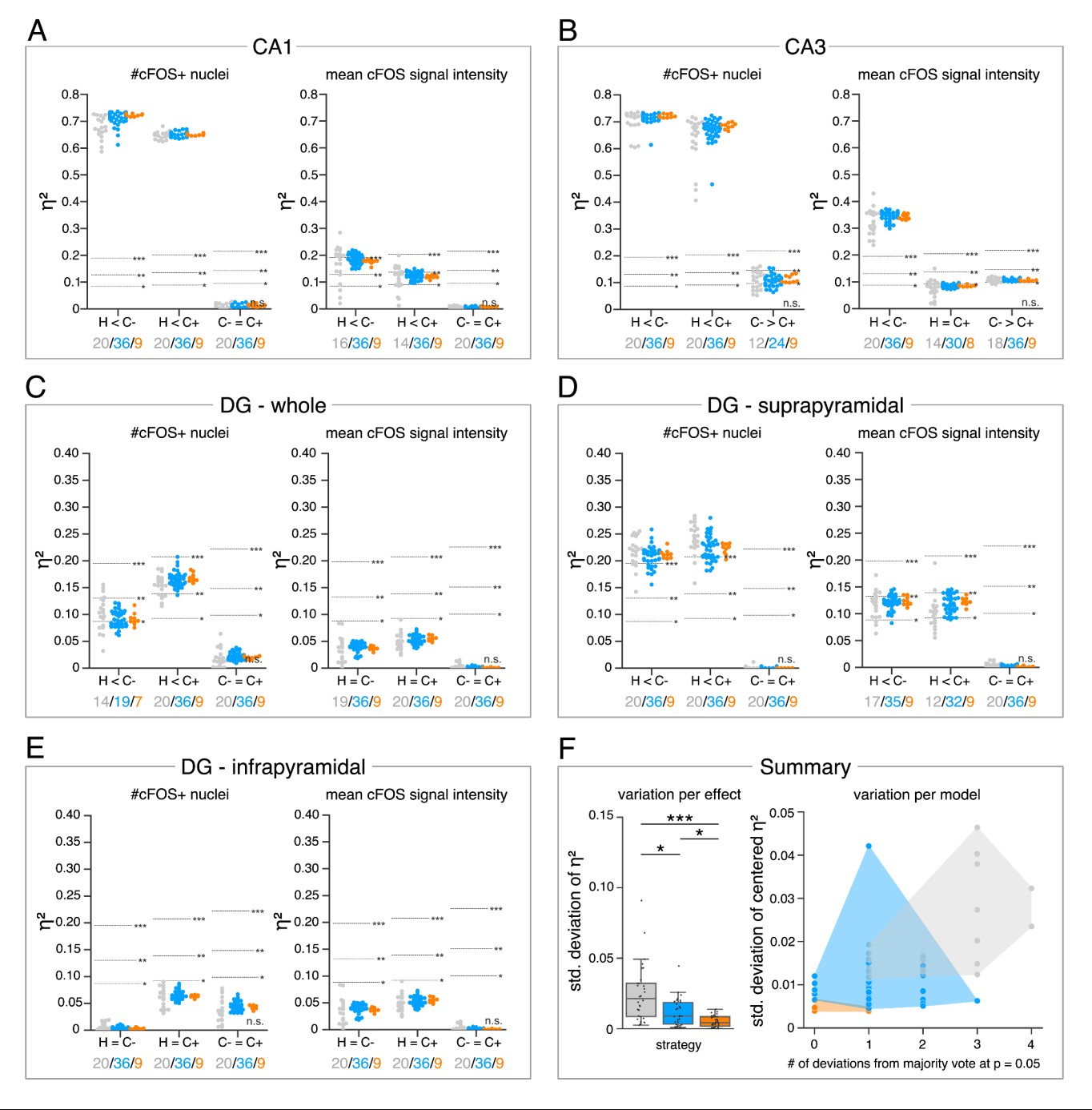

**Figure 4.** Consensus ensembles significantly increase reliability of bioimage analysis results. (A–E) Single data points represent the calculated effect sizes for each pairwise comparison of all individual bioimage analyses for each DL-based strategy (gray: expert models, blue: consensus models, orange: consensus ensembles) in indicated hippocampal subregions. Three horizontal lines separate four significance intervals (n.s.: not significant, *: $0.05 \geq p > 0.01$, **: $0.01 \geq p > 0.001$, ***: $p \leq 0.001$ after Bonferroni correction for multiple comparisons). The quantity of analyses of each strategy that report the respective statistical result of the indicated pairwise comparison (effect, x-axis) at a level of $p \leq 0.05$ are given below each pairwise comparison in the corresponding color coding. In total, we performed all analyses with: $N_{expert\ models} = 20$, $N_{consensus\ models} = 36$, $N_{consensus\ ensembles} = 9$. Number of analyzed mice (N) for all analyzed subregions: $N_H = 7$, $N_{C-} = 7$, $N_{C+} = 6$. Numbers of analyzed images (n) are given for each analyzed subregion. Source files including source data and statistical data are available in *Figure 4—source data 1*. (A) Analyses of cFOS-positive nuclei in stratum pyramidale of CA1. $n_H = 36$, $n_{C-} = 32$, $n_{C+} = 28$. (B) Analyses of cFOS-positive nuclei in stratum pyramidale of CA3. $n_H = 35$, $n_{C-} = 31$, $n_{C+} = 28$. (C) Analyses of cFOS-positive nuclei in the granule cell layer of the whole DG. $n_H = 35$, $n_{C-} = 31$, $n_{C+} = 27$. (D) Analyses of cFOS-positive nuclei in the granule cell layer of the suprapyramidal blade of the DG. $n_H = 35$, $n_{C-} = 31$, $n_{C+} = 27$. (E) Analyses of cFOS-positive nuclei in the granule cell layer of the

*Figure 4 continued on next page*

*Figure 4 continued*

infrapyramidal blade of the DG. $n_H$ = 35, $n_{C-}$ = 31, $n_{C+}$ = 27. (F) Reliability of bioimage analysis results are assessed as *variation per effect* (left side) and *variation per model* (right side). For the *variation per effect*, single data points represent the standard deviation of reported effect sizes ($\eta^2$), calculated within each DL-based strategy for each of the 30 pairwise comparisons. Consensus ensembles show significantly lower standard (std.) deviations of $\eta^2$ per pairwise comparison compared to alternative strategies ($X^2$(2) = 26.472, p<0.001, $N_{effects}$ = 30, Kruskal-Wallis ANOVA followed by pairwise Mann-Whitney tests with Bonferroni correction, *p<0.05, ***p<0.001). For the *variation per model*, the standard deviation of centered $\eta^2$ across all pairwise comparisons was calculated for each individual model and ensemble (y-axis). In addition, the number of deviations from the congruent majority vote (at p ≤ 0.05 after Bonferroni correction for multiple comparisons) were determined for each individual model and ensemble across all pairwise comparisons (x-axis). Visualizing the interaction of both measures for each model or model ensemble individually reveals that consensus ensembles show the highest reliability of all three DL-based strategies. The statistical data for the for variation per effect is available in *Figure 4—source data 2*. The online version of this article includes the following source data for figure 4:

**Source data 1.** Source files for the analysis of cFOS positive nuclei in the hippocampal subregions.
**Source data 2.** Statistical data for the variation per effect.

The meta analysis discloses a context-dependent increase of cFOS in almost all analyzed hippocampal regions (*Figure 4A–D*), except for the infrapyramidal blade of the dentate gyrus (*Figure 4E*). Notably, the majority votes of all three strategies at a significance level of p ≤ 0.05 (after Bonferroni correction for multiple comparisons) are identical for each pairwise comparison (*Figure 4A–E*). However, the results can vary between individual models or ensembles (*Figure 4A–E*).

In order to assess the reliability of bioimage analysis results of the individual strategies, we further examined the variation per effect and variation per model in *Figure 4F*. For the variation per effect, we calculated the standard deviation of reported effect sizes within each strategy for every pairwise comparison (effect). This confirmed the visual impression from *Figure 4A–E* as the consensus ensembles yield a significantly lower standard deviation compared to both alternative strategies (*Figure 4F*). To illustrate the variation per model, we show the interaction between the number of biological effects that the corresponding model (or ensemble) reported differently compared to the congruent majority votes versus the standard deviation of its centered effect sizes across all 30 analyzed effects. This analysis shows that no expert model detected all biological effects in the microscopy images that were defined by the majority votes of all models. This is in stark contrast to the consistency of effect interpretation across the consensus ensembles (*Figure 4F*).

Consequently, we conclude that the consensus ensemble strategy is best suited to satisfy the bioimaging standards for objectivity, reliability, and validity.

## Applicability of consensus ensemble strategy for the bioimage analysis of external data sets

Bioimage analysis of fluorescent labels comes with a huge variability in terms of investigated model organisms, analyzed fluorescent features and applied image acquisition techniques (*Meijering et al., 2016*). In order to assess our consensus ensemble strategy across these varying parameters, we tested it on four external datasets that were created in a fully independent manner and according to individual protocols (*Lab-Mue, Lab-Inns1, Lab-Inns2,* and *Lab-Wue2*; see Materials and methods and *Figure 5—source data 2*). Due to limited dataset sizes, the lab-specific training datasets consisted of just five microscopy images each and the corresponding est. GT based on the annotations from multiple experts. In the biomedical research field, the limited availability of training data is a common problem when training DL algorithms. For this reason, extensive data augmentation and regularization techniques, as well as transfer learning strategies are widely used to cope with small datasets (*Ronneberger et al., 2015*; *Christiansen et al., 2018*; *Falk et al., 2019*). Transfer learning is a technique that enables DL models to reuse the image feature representations learned on another source, such as a task (e.g. image segmentation) or a domain (e.g. the fluorescent feature, here cFOS-positive nuclei). This is particularly advantageous when applied to a task or domain where limited training data is available (*Yosinski et al., 2014*; *Oquab et al., 2014*). Moreover, transfer learning might be used to reduce observer variability and to increase feature annotation objectivity (*Bayramoglu and Heikkilä, 2016*). There are typically two ways to implement transfer learning for DL models, either by fine-tuning or by freezing features (i.e. model weights) (*Yosinski et al., 2014*). The latter approach, if applied to the same task (e.g. image segmentation), does not require any

further model training. These *out-of-the-box* models reduce time and hardware requirements and may further increase objectivity of image analysis, by altogether excluding the need for any additional manual input.

Consequently, we hypothesized that transfer learning from pretrained model ensembles would substantially reduce the training efforts (*Falk et al., 2019*) and might even increase objectivity of bioimage analysis. To test this, we followed three different initialization variants of the consensus ensemble strategy (*Figure 5A*). In addition to starting the training of DL models with randomly initialized weights (*Figure 5A - from scratch*), we reused the consensus ensemble weights from the previous evaluation (*Lab-Wue1*) by means of fine-tuning (*Figure 5A - fine-tuned*) and freezing of all model layers (*Figure 5A - frozen*). Although no training of the *frozen* model is required, we tested and evaluated the performance of *frozen* models to ensure their validity. After performing the similarity analysis, we compared the full bioimage analyses, including quantification and hypothesis testing, of the different initialization variants. Finally, to establish a notion of external validity, we also compared these results with the manually and independently performed bioimage analysis of a lab-specific expert (*Figure 5*, *Figure 5—figure supplement 1*, and *Figure 5—figure supplement 2*).

## Dataset characteristics

The first dataset (*Lab-Mue*) represents very similar image parameters compared to our original *Lab-Wue1* dataset (*Figure 5C - Lab-Mue* and *Figure 5—source data 2*). Mice experienced restraint stress and subsequent Pavlovian fear conditioning (cue-conditioning, tone-footshock association) and the number of cFOS-positive cells in the paraventricular thalamus (PVT) was compared between early (eRet) and late (lRET) phases of fear memory retrieval. In the context of transfer learning, this dataset originates from a very similar domain and requires the same task (image segmentation). Another two external datasets are focused on the quantification of cFOS abundance (similar domain), albeit showing less similarity in image parameters to our initial dataset (*Figure 5—figure supplement 1*, *Figure 5—figure supplement 2* and *Figure 5—source data 2*). In *Lab-Inns1*, mice underwent Pavlovian fear conditioning and extinction in the same context. The image dataset of *Lab-Inns2* shows cFOS immunoreactivity in the infralimbic cortex (IL) following fear renewal, meaning return of extinguished fear in a context different from the extinction training context. Since heterogeneity in this behavioral response was observed, mice were classified as responders (Resp) or non-responders (nResp), based on freezing responses (see Materials and methods). The image dataset of *Lab-Wue2* shows the least similarity of image parameters to the dataset of *Lab-Wue1*. This dataset represents another commonly used model organism in neurobiology, the zebrafish. Here, cell bodies of specific neurons (GABAergic neurons) instead of nuclei were fluorescently labeled (*Figure 5C - Lab-Wue2* and *Figure 5—source data 2*). Hence, this dataset originates from a different domain but was acquired using the same technique.

## Similarity analysis

As only limited training data was available, we executed the similarity analysis for all external datasets by means of a *k-fold cross-validation*. We observed that the inter-rater variability differed between laboratories and different experts but remained comparable as previously for *Lab-Wue1* (*Figure 5D*, *Figure 5—figure supplement 3*, and *Figure 5—figure supplement 4*.) Both *from scratch* and *fine-tuned* initiation variants resulted in individual consensus models that reached human expert level performance (*Figure 5D*, *Figure 5—figure supplement 1*, *Figure 5—figure supplement 2*). However, models adapted from pretrained weights yielded a higher validity in terms of similarity to the estimated ground truth. They either exceeded the maximal $M_{\text{F1score}}$ reached by *from scratch* models (*Figure 5D - Lab-Mue*, *Figure 5—figure supplement 1*, *Figure 5—figure supplement 2*) or reached them after less training iterations (*Figure 5D - Lab-Wue2*). As expected, the performance of *frozen Lab-Wue1*-specific consensus models was highly dependent on the image similarity between the original and the new dataset. Consequently, the *out-of-the-box* segmentation performance of the *frozen Lab-Wue1* models was very poor on dissimilar images (*Figure 5D - Lab-Wue2*), but we found it to be on par with human experts and adapted models on images that are highly similar to the original dataset (*Figure 5D - Lab-Mue* - very similar domain and the same task).

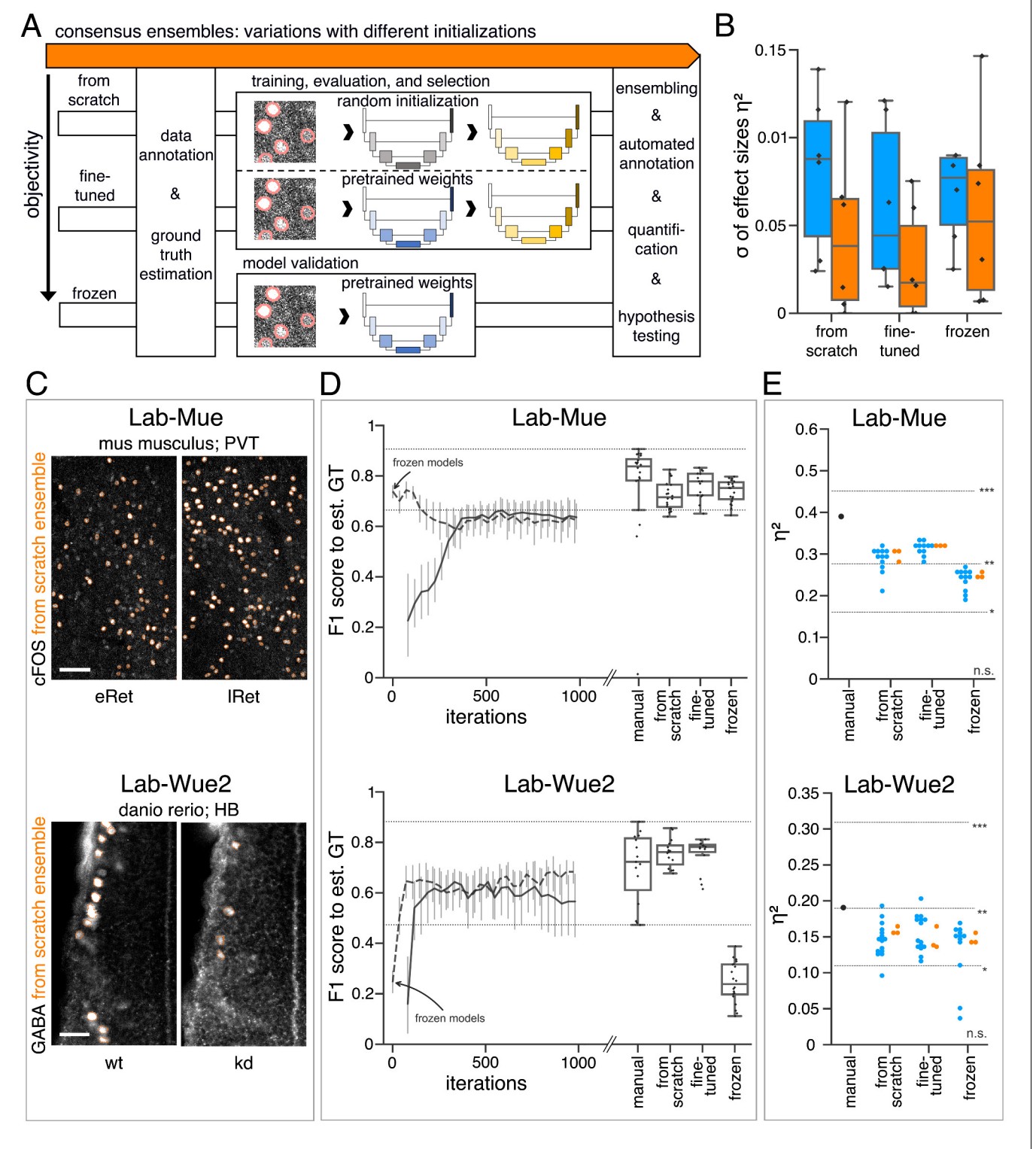

**Figure 5.** Consensus ensembles for DL-based feature annotation in external bioimage data sets. (**A**) Schematic overview depicting three initialization variants for creating consensus ensembles on new datasets. Data annotation by multiple human experts and subsequent ground truth estimation are required for all three initialization variants. In the *from scratch* variant, a U-Net model with random initialized weights is trained on pairs of microscopy images and estimated ground truth annotations. This variant was used to create consensus ensembles for the initial *Lab-Wue1* dataset. Alternatively, the same training dataset can be used to adapt a U-Net model with pretrained weights by means of transfer-learning (*fine-tuned*). In both variants, models are evaluated and selected on base of a validation set after model training. In a third variant, U-Net models with pretrained weights can be

*Figure 5 continued on next page*

Figure 5 continued

evaluated directly on a validation dataset, without further training (*frozen*). In all three variants, consensus ensembles of the respective models are then used for bioimage analysis. (B) Overall reliability of bioimage analysis results of each variant assessed as variation per effect. In all three strategies, consensus ensembles (orange) showed lower standard deviations than consensus models (blue). The *frozen* results need to be considered with caution as they are based on models that did not meet the selection criterion (see *Figure 5—source data 3*). $N_{pairwise\ comparisons} = 6$; $N_{consensus\ models} = 15$, and $N_{consensus\ ensembles} = 3$ for each variant. (C–E) Detailed comparison of the two external datasets with highest (*Lab-Mue*) and lowest (*Lab-Wue2*) similarity to *Lab-Wue1*. (C) Representative microscopy images. Orange: representative annotations of a lab-specific *from scratch* consensus ensemble. PVT: paraventricular nucleus of thalamus, eRet: early retrieval, lRet: late retrieval, HB: hindbrain, wt: wildtype, kd: *gad1b* knock-down. Scale bars: *Lab-Mue* 100 µm and *Lab-Wue2* 6 µm. (D) Mean $M_{F1score}$ of *from scratch* (solid line) and *fine-tuned* (dashed line) consensus models on the validation dataset over the course of training (iterations). Mean $M_{F1score}$ of *frozen* consensus models are indicated with arrows. Box plots show the $M_{F1score}$ among the annotations of human experts as reference and the mean $M_{F1score}$ of selected consensus models. Two dotted horizontal lines mark the whisker ends of the $M_{F1score}$ among the human expert annotations. (E) Effect sizes of all individual bioimage analyses (black: manual experts, blue: consensus models, orange: consensus ensembles). Three horizontal lines separate the significance intervals (n.s.: not significant, *: $0.05 \geq p > 0.01$, **$0.01 \geq p > 0.001$, ***$p \leq 0.001$ with Mann-Whitney-U tests). *Lab-Mue*: $N_{consensus\ ensembles} = 3$ for all initialization variants; $N_{from\ scratch/fine-tuned\ consensus\ models} = 12$ (for each ensemble, 4/5 trained models per ensemble met the selection criterion), $N_{frozen\ consensus\ models} = 12$ (for each ensemble, 4/4 models per ensemble did not meet the selection criterion). $N_{eRet} = 4$, $N_{lRet} = 4$; $n_{eRet} = 12$, $n_{lRet} = 11$. *Lab-Wue2*: $N_{consensus\ ensembles} = 3$ for each initialization variant; $N_{from\ scratch/fine-tuned\ consensus\ models} = 15$ (for each ensemble, 5/5 trained models per ensemble met the selection criterion), $N_{frozen\ consensus\ models} = 12$ (for each ensemble, 4/4 models per ensemble did not meet the selection criterion). $N_{wt} = 5$, $N_{kd} = 4$, $n_{wt} = 20$, $n_{kd} = 15$. Source files of all statistical analyses (including *Figure 5—figure supplement 2* and *Figure 5—figure supplement 1*) are available in *Figure 5—source data 1*. Information on all bioimage datasets (e.g. the number of images, image resolution, imaging techniques, etc.) are available in *Figure 5—source data 2*. Source files on model performance and selection are available in (*Figure 5—source data 3*).

The online version of this article includes the following source data and figure supplement(s) for figure 5:

**Source data 1.** Statistical data for *Lab-Mue*, *Lab-Wue2*, *Lab-Inns1*, and *Lab-Inns2*.
**Source data 2.** Characteristics of all five bioimage datasets.
**Source data 3.** Model performance with selection criterion for *Lab-Mue*, *Lab-Wue2*, *Lab-Inns1*, and *Lab-Inns2*.
**Figure supplement 1.** Performance of consensus ensembles on feature annotation in image dataset Lab-Inns01.
**Figure supplement 2.** Performance of consensus ensembles on fluorescent feature annotation in image dataset Lab-Inns02.
**Figure supplement 3.** Expert similarity across all datasets: F1 scores.
**Figure supplement 4.** Expert similarity across all datasets: mean IoU.
**Figure supplement 5.** Reliability of the consensus approaches across different datasets.

## Bioimage analysis results

To further strengthen the validity of our workflow, we compared all DL-based bioimage analyses to the manual analysis of a human expert from the individual laboratory (*Figure 5E*, *Figure 5—figure supplement 1*, *Figure 5—figure supplement 2*, and *Table 1*).

For *Lab-Mue*, the bioimage analyses of all DL-based approaches, including the *frozen* consensus models and ensembles pretrained on *Lab-Wue1*, revealed a significantly higher number of cFOS-positive cells in the PVT of mice 24 hr after fear conditioning (lRET), which was confirmed by the manual expert analysis (*Figure 5E - Lab-Mue*, *Table 1*). Yet again, the formation of model ensembles increased the reproducibility of results by yielding less or almost no variation in the effect sizes (*Figure 5E - Lab-Mue*).

The manual expert analysis of the *Lab-Inns1* dataset revealed a significantly higher number of cFOS-positive nuclei in the basolateral amygdala (BLA) after extinction of a previously learned fear, which was also reliably detected by all consensus ensembles, regardless of initiation variant (*Figure 5—figure supplement 1*, *Table 1*). However, this significant difference was only present in the analyses of most individual consensus models, both *from scratch* and *fine-tuned* (*Figure 5—figure supplement 1*). Again, this could be attributed to a higher variability between the effect sizes of individual models, compared to a higher homogeneity among ensembles (*Figure 5—figure supplement 1*).

For *Lab-Inns2*, the manual expert analysis as well as all DL-based approaches that were adapted to the *Lab-Inns2* dataset show increased numbers of cFOS-positive cells in the infralimbic cortex of L-DOPA/MS-275 responders (Resp) compared to control (Sal) mice (*Figure 5—figure supplement 2*, *Table 1*). However, in L-DOPA/MS-275 non-responders (nResp), we did not observe a significant increase of cFOS-positive nuclei (*Figure 5—figure supplement 2*, *Table 1*). Furthermore, the high effect sizes of the comparison between L-DOPA/MS-275 responders and non-responders further indicate that the differences observed in the behavioral responses of Resp and nResp mice were

**Table 1.** Bioimage analyses results of external datasets.

Data are based either on manual analysis or on annotations by a consensus ensemble. The results are given for the individual consensus ensemble initialization variants (*from scratch, fine-tuned*). p-Values of *Lab-Inns2* are corrected for multiple comparisons using Bonferroni correction. $\mu_1$: mean group 1, $\mu_2$: mean group 2, U: U-statistic, eRet: early retrieval, lRet: late retrieval, Ctrl: control, Ext: extinction, Sal: saline, Res: L-DOPA/MS-275 responder, nRes: L-DOPA/MS-275 non-responder, wt: wildtype, kd: *gad1b* knock-down.

| Lab | Groups | Initialization variant | $\mu_1$ | $\mu_2$ | U | Significance level (p) | $\eta^2$ |
|---|---|---|---|---|---|---|---|
| Mue | eRet ~ lRet | Manual | 1.00 | 1.65 | 19.0 | ** (0.002) | 0.39 |
| | | From scratch | 1.00 | 1.70 | 25.0 | ** (0.007) | 0.31 |
| | | Fine-tuned | 1.00 | 1.68 | 24.0 | ** (0.006) | 0.32 |
| Inns1 | Ctrl ~ Ext | Manual | 1.00 | 3.92 | 10.0 | ** (0.005) | 0.43 |
| | | From scratch | 1.00 | 2.26 | 13.0 | * (0.010) | 0.35 |
| | | Fine-tuned | 1.00 | 1.85 | 14.0 | * (0.013) | 0.33 |
| Inns2 | Sal ~ Resp | Manual | 1.00 | 1.83 | 5.0 | ** (0.002) | 0.59 |
| | | From scratch | 1.00 | 1.96 | 0.0 | *** (<0.001) | 0.71 |
| | | Fine-tuned | 1.00 | 2.07 | 0.0 | *** (<0.001) | 0.71 |
| | Sal ~ nResp | Manual | 1.00 | 1.05 | 27.0 | n.s. (1.000) | 0.00 |
| | | From scratch | 1.00 | 1.63 | 8.0 | n.s. (0.130) | 0.29 |
| | | Fine-tuned | 1.00 | 1.42 | 12.0 | n.s. (0.377) | 0.16 |
| | Res ~ nRes | Manual | 1.83 | 1.05 | 42.0 | n.s. (0.130) | 0.29 |
| | | From scratch | 1.96 | 1.63 | 41.0 | n.s. (0.173) | 0.26 |
| | | Fine-tuned | 2.07 | 1.42 | 42.0 | n.s. (0.130) | 0.29 |
| Wue2 | wt ~ kd | Manual | 1.00 | 0.28 | 227.5 | * (0.010) | 0.19 |
| | | From scratch | 1.00 | 0.45 | 220.0 | * (0.021) | 0.16 |
| | | Fine-tuned | 1.00 | 0.37 | 216.0 | * (0.029) | 0.14 |

also reflected in the abundance of cFOS in the infralimbic cortex (*Figure 5—figure supplement 2*, *Table 1*).

Manual expert analysis of the fourth external dataset revealed a significantly lower amount of GABA-positive somata in *gad1b* knock-down zebrafish, compared to wildtypes (*Figure 5E - Lab-Wue2*, *Table 1*). Again, this effect was reliably detected by all deep-learning-based approaches that included training on the *Lab-Wue2*-specific training dataset and the effect sizes of ensembles showed less variability (*Figure 5E - Lab-Wue2*). Despite its poor segmentation performance and hence, poor validity, this effect was also present in the bioimage analysis of the *frozen* consensus models and ensembles pretrained on *Lab-Wue1* (*Figure 5E - Lab-Wue2*).

As with our initial dataset, we assessed reliability by calculating the variation per effect as the standard deviation of the reported effect sizes within each group and pooled these results across all external datasets. Consistent with the higher reliability of *from scratch* and *fine-tuned* ensemble annotations (*Figure 5—figure supplement 5*), this analysis shows that the formation of model ensembles reduced the variation per effect in both variants, compared to the respective individual models (*Figure 5B*). The *frozen* models and ensembles exhibit a similar effect, but need to be considered with caution as they are based on models that did not meet the selection criterion (reliably performing on par with human experts; see 7.10.4 - Training, evaluation and model selection for a detailed explanation).

In summary, we assessed the reproducibility of our consensus ensemble strategy by using four external datasets. These datasets were acquired with different image acquisition techniques, investigate two common model organisms, and analyze the two main cellular compartments (nuclei and somata) at varying resolutions (*Figure 5—source data 2*). In-line with the results obtained on our initial dataset, we observed an increased reproducibility for the consensus ensembles compared to individual consensus models after training on all four external datasets (*Figure 5B*).

Moreover, our data also suggests that pretrained consensus models can even be deployed *out-of-the-box*, but only when carefully validated. Thus, sharing pretrained model weights across different laboratories reduces lab-specific biases within the bioimage analysis and may further increase objectivity and validity.

Ultimately, we conclude that our proposed ensemble consensus workflow is reproducible for different datasets and laboratories and increases objectivity, reliability, and validity of DL-based bioimage analyses.

## Discussion

The present study contributes to bridging the gap between 'methods' and 'biology' oriented studies in image feature analysis (*Meijering et al., 2016*). We explored the potentials and limitations of DL models utilizing the general quality criteria for quantitative research: objectivity, reliability, and validity. Thereby, we put forward an effective but easily implementable strategy that aims to establish reproducible, DL-based bioimage analysis within the life science community.

The number of DL-based tools for bioimage annotations and their accessibility for non-AI specialists is gradually increasing (*McQuin et al., 2018*; *Haberl et al., 2018*; *Falk et al., 2019*). DL models can hold advantages over conventional algorithms (*Caicedo et al., 2019*) and have the potential to be commonly used for bioimage analysis tasks throughout the life sciences. Usually, the performance of new bioimage analysis tools or methods is assessed by means of similarity measures to a certain ground truth (*Ronneberger et al., 2015*; *McQuin et al., 2018*; *Haberl et al., 2018*; *Falk et al., 2019*; *Caicedo et al., 2019*). However, this is rarely sufficient to establish trust in the use of DL models for bioimage analysis, as the vast amount of parameters and flexibility to adapt DL models to virtually any task renders them prone to internalize unintended, but subjective human biases (*von Chamier et al., 2019*). This is particularly true in the case of fluorescent feature analysis in bioimage datasets, as an objective ground truth is not available. In conjunction with the stochastic training process, this is a very critical point, because it holds the potential for intended or unintended tampering similar to p-hacking (*Head et al., 2015*), for example by training DL models until non-significant results become significant.

To investigate the effects of DL-based strategies on the bioimage analysis of fluorescent features, we acquired a typical bioimage dataset (*Lab-Wue1*) and five experts manually annotated corresponding ROIs (here cFOS-positive nuclei) in a representative subset of images. Then, we tested three DL-based strategies for automatized feature segmentation. DL models were either trained on the manual annotations of a single expert (expert models) or on the input of multiple experts pooled by ground truth estimation (consensus models). In addition, we formed ensembles of consensus models (consensus ensembles).

### Similarity analysis of fluorescent feature annotation

In accordance with previous studies, similarity analyses revealed a substantial level of inter-rater variability in the heuristic annotations of the single experts (*Schmitz et al., 1999*; *Collier et al., 2003*; *Niedworok et al., 2016*). Furthermore, we confirmed the concerns already put forward by others (*Falk et al., 2019*; *von Chamier et al., 2019*) that training of DL models solely on the input of a single human expert imposes a high risk of incorporating an individual human bias into the trained models. We therefore conclude that models trained on single expert annotations resemble an automation of manual image annotation, but cannot remove subjective biases from bioimage analyses. Importantly, only consensus ensembles led to a coincident significant increase also in the reliability and validity of fluorescent feature annotations. Our analyses also show that annotations of multiple experts are imperative for two reasons: first, they mitigate or even eliminate the bias of expert-specific annotations and, second, are essential for the assessment of the model performance.

### Reproducibility and validity of bioimage analyses

Our bioimage dataset from *Lab-Wue1* enabled us to look at the impact of different DL-based strategies on the results of bioimage analyses. This revealed a striking model-to-model variability as the main factor impairing the reproducibility of DL-based bioimage analyses. Convincingly, the majority votes for each effect were identical for all three strategies. However, the variance within the reported effect sizes differed significantly for each strategy. This entailed, for example, that no

expert model was in full agreement with the congruent majority votes. On the contrary, consensus ensembles detected all effects with significantly higher reliability. Thus, our data indicates that bioimage analysis performed with a consensus ensemble significantly reduces the risk of obtaining irreproducible results.

## Evaluation of consensus ensembles on external datasets

We then tested our consensus ensemble approach and three initialization variants on four external datasets with limited training data and varying similarities in terms of image parameters to our original dataset (*Lab-Wue1*). In line with previous studies on transfer learning, we demonstrate that the adaptation of models from pretrained weights to new, yet similar data requires less training iterations, compared to the training of models *from scratch* (*Falk et al., 2019*). We extend these analyses and show that the reliability of *fine-tuned* ensembles was at least equivalent to *from scratch* ensembles, if not higher. Furthermore, we also provide initial evidence that pretrained ensembles can be used even without any adaptation, if task similarity is sufficiently high. Our data suggest that this component in the analysis pipeline could further increase the objectivity of bioimage analyses.

## Potentials of open-source pretrained consensus ensemble libraries

Sharing model weights from validated models in open-source libraries, similarly to *TensorFlow Hub* (https://www.tensorflow.org/hub) or *PyTorch Hub* (https://pytorch.org/hub/), offers a great opportunity to provide annotation experience across labs in an open science community. In this study, for instance, we used the nuclear label of cFOS, an activity-dependent transcription factor, as fluorescent feature of interest. This label is in its signature indistinguishable from a variety of other fluorescent labels, like those of transcription factors (CREB, phospho-CREB, Pax6, NeuroG2, or Brain3a), cell division markers (phospho-histone H3), apoptosis markers (Caspase-3), and multiple others. Similarly to the pretrained and shared models of *Falk et al., 2019*, we surmise that the learned feature representations (i.e. model weights) of our cFOS consensus ensembles may serve as a good initialization for models that aim at performing nucleosomatic fluorescent label segmentation in brain slices.

In line with the results of the *Kaggle Data Science Bowl 2018* (*Caicedo et al., 2019*), however, our findings indicate that a model adapted to a specific data set usually outperforms a general model trained on different datasets from different domains. To use and share frozen *out-of-the-box* models across the science community, we therefore need to create a well-documented library that contains validated model weights for each specific task and domain (e.g. for each organism, marker type, image resolution, etc.). In conjunction with data repositories, this would also allow retrospective data analysis of prior studies.

In summary, open-source model libraries may contribute to a better reproducibility of scientific experiments (*Fanelli, 2018*) by improving the objectivity in bioimage analyses, by offering openness to analysis criteria, and by sharing pretrained models for (re-)evaluation.

## Limitations

This paper describes a blueprint for the evaluation of DL models in biomedical imaging. Therefore, some of our methodological decisions were shaped by standardization considerations concerning the future deployment in bioimage analysis pipelines.

The project was triggered by segmentation tasks for fluorescent labels (cFOS) in the cell nucleus. These are rather simple features, and we could readily annotate data from different labs, which facilitated the evaluation. However, this limits the generalizability to more complex image segmentation tasks, where training data annotation is slow and tedious. In particular, human perceptive capabilities for richer graphical features, such as area, volume, or density, is much worse than for regular, linear image features (*Cleveland and McGill, 1985*; *Feldman-Stewart et al., 2000*). A case in point is the annotation of images showing ramified neurons or astrocytes. Such tasks would cause an enormous workload rendering complete human annotation virtually impossible. In this respect, we concur with prior research asserting that DL models based on human annotations will not be an option in these settings (*Driscoll et al., 2019*).

The characteristics of our examined strategies are based on best practices in the field of DL and derived from extant literature (*Meijering et al., 2016*; *Falk et al., 2019*; *Caicedo et al., 2019*). The

focus on the U-Net model architecture (*Ronneberger et al., 2015*) is a direct consequence of this standardization idea. Yet, it is also an important limitation of our study. Unlike more conventional studies that introduce a new method and provide a comprehensive performance comparison to the state of the art, we rely on U-net as the widely studied de facto standard for biomedical image segmentation purposes (*McQuin et al., 2018*; *Falk et al., 2019*; *Caicedo et al., 2019*). Similarly, we chose to use STAPLE (*Warfield et al., 2004*) as the benchmark procedure for ground truth estimation. Thereby, we forwent considering alternatives and variants (*Lampert et al., 2016*). In addition, we tried different ways to incorporate the single expert annotations into one DL model. For instance, we followed the approach of *Guan et al., 2018* by modeling individual experts in a multi-head DL model instead of pooling them in the first place. However, we decided to discard the approach as our tests did not improve the results but increased complexity.

### Accessibility of our workflow and pretrained consensus ensembles

To enable other researchers to easily access, to interact with, and to reproduce our results and to share our trained models, we provide an open-source *Python* library that is easily accessible for both local installation or cloud-based deployment.

With *Jupyter Notebooks* becoming the computational notebook of choice for data scientists (*Perkel, 2018*), we also implemented a training pipeline for non-AI experts in a *Jupyter Notebook* optimized for Google Colab, providing free access to the required computational resources (e.g., GPUs and TPUs). In summary, we recommend to use the annotations of multiple human experts to train and evaluate DL consensus model ensembles. In such a way, DL can be used to increase the objectivity, reliability, and validity of bioimage analyses and pave the way for higher reproducibility in science.

# Materials and methods

**Key resources table**

| Reagent type (species) or resource | Designation | Source or reference | Identifiers | Additional information |
|---|---|---|---|---|
| Genetic reagent (*Mus musculus*, male) | C57BL/6J | Charles River | Cat# CRL:027; RRID:IMSR_CRL:27 | Lab-Mue; Lab-Inns1 |
| Genetic reagent (*Mus musculus*, male) | C57BL/6J | Jackson Laboratory | Cat# JAX:000664; RRID:IMSR_JAX:000664 | Lab-Wue1 |
| Genetic reagent (*Mus musculus*, male) | 129S1/SvlmJ (S1) | Charles River | RRID:MGI:5658424 | Lab-Inns2 |
| Genetic reagent (*Danio rerio*) | AB/AB | European Zebrafish Resource Center | | Lab-Wue2 |
| Antibody | Anti-cFOS (rabbit polyclonal) | Santa Cruz | Cat# sc-52; RRID:AB_2106783 | Lab-Mue (1:500); Lab-Inns2 (1:1,000) |
| Antibody | Anti-cFOS (rabbit polyclonal) | Millipore | Cat# PC38; RRID:AB_2106755 | Lab-Inns1 (1:20,000) |
| Antibody | anti-cFOS (rabbit polyclonal) | Synaptic Systems | Cat# 226003; RRID:AB_2231974 | Lab-Wue1 (1:10,000) |
| Antibody | Anti-GABA (rabbit polyclonal) | Sigma-Aldrich | Cat#A2025; RRID:AB_477652 | Lab-Wue2 (1:400) |
| Antibody | Anti-NeuN (guinea-pig polyclonal) | Synaptic Systems | Cat# 266004; RRID:AB_2619988 | Lab-Wue1 (1:400) |
| Antibody | Anti-Parvalbumin (mouse monoclonal) | Sigma-Aldrich | Cat# P3088; RRID:AB_477329 | Lab-Inns1 (1:2,500) |
| Antibody | Anti-Parvalbumin (mouse monoclonal) | Swant | Cat# PV235; RRID:AB_10000343 | Lab-Wue1 (1:5,000) |
| Software, algorithm | ImageJ | Fiji www.fiji.sc/ | RRID:SCR_002285 | Lab-Mue; Lab-Inns2; Lab-Wue1; Lab-Wue2 |

*Continued on next page*

*Continued*

| Reagent type (species) or resource | Designation | Source or reference | Identifiers | Additional information |
|---|---|---|---|---|
| Software, algorithm | Improvision Openlab software | Perkin Elmer www.perkinelmer.com/ pages/020/cellularimaging/ products/openlab.xhtml | RRID:SCR_012158 | Lab-Inns1, Version 5.5.0 |
| Software, algorithm | GraphPad Prism software | GraphPad Prism www.graphpad.com/ scientific-software/prism/ | RRID:SCR_015807 | Lab-Inns1, Version 7.0 |
| Software, algorithm | CellSens Dimension Desktop software | Olympus www.olympus-lifescience.com/ en/software/cellsens/ | RRID:SCR_016238 | Lab-Inns2, Version 1.9 |
| Software, algorithm | Fluoview FV10-ASW | Olympus www.photonics.com/ Product.aspx?PRID=47380 | RRID:SCR_014215 | Lab-Wue1 |
| Software, algorithm | Tensorflow | www.tensorflow.org, *Abadi et al., 2016* | RRID:SCR_016345 | |
| Software, algorithm | Keras | www.keras.io, *Chollet, 2015* | | |
| Software, algorithm | Imagej | www.imagej.net/, *Rueden et al., 2017* | RRID:SCR_003070 | |
| Software, algorithm | SciPy | www.scipy.org, *Jones et al., 2001* | RRID:SCR_008058 | |
| Software, algorithm | scikit-learn | www.scikit-learn.org/, *Pedregosa et al., 2011* | | |
| Software, algorithm | scikit-image | www.scikit-image.org/, *van der Walt et al., 2014* | | |
| Software, algorithm | Pingouin | https://pingouin-stats.org/, *Vallat, 2018* | | |
| Software, algorithm | simpleITK | www.simpleitk.org/, *Lowekamp et al., 2013* | | |

Data sets regarding animal behavior, immunofluorescence analysis and image acquisition were performed in five independent laboratories using lab-specific protocols. Experiments were not planned together to ensure the individual character of the datasets. We refer to the lab-specific protocols as follows:

- *Lab-Mue*: Institute of Physiology I, University of Münster, Germany
- *Lab-Inns1*: Department of Pharmacology, Medical University of Innsbruck, Austria
- *Lab-Inns2*: Department of Pharmacology and Toxicology, Institute of Pharmacy and Center for Molecular Biosciences Innsbruck, University of Innsbruck
- *Lab-Wue1*: Institute of Clinical Neurobiology, University Hospital, Würzburg, Germany
- *Lab-Wue2*: Department of Child and Adolescent Psychiatry, Center of Mental Health, University Hospital of Würzburg, Würzburg, Germany

## Contacts for reagent and resource sharing

Further information and requests for resources and reagents should be directed to and will be fulfilled by the lead contact, Robert Blum (Blum_R@ukw.de). Requests regarding the machine learning model and infrastructure should be directed to Christoph M. Flath (christoph.flath@uni-wuerzburg. de).

## Experimental models

### Mice

#### Lab-Mue

Male C57Bl/6J mice (Charles River, Sulzfeld, Germany) were kept on a 12hr-light-dark cycle and had access to food and water ad libitum. No more than five and no less than two mice were kept in a

cage. Experimental animals of an age of 9–10 weeks were single housed for 1 week before the experiments started. All animal experiments were carried out in accordance with European regulations on animal experimentation and protocols were approved by the local authorities (Landesamt für Natur, Umwelt und Verbraucherschutz Nordrhein-Westfalen).

### Lab-Inns1
Experiments were performed in adult, male C57Bl/6NCrl mice (Charles River, Sulzfeld, Germany) at least 10–12 weeks old, during the light phase of the light/dark cycle. They were bred in the Department of Pharmacology at the Medical University Innsbruck, Austria in Sealsafe IVC cages (1284L Eurostandard Type II L: 365 × 207×140 mm, floor area cm$^2$ 530, Tecniplast Deutschland GmbH, HohenpeiÃŸenberg, Germany). Mice were housed in groups of three to five animals under standard laboratory conditions (12 hr/12 hr light/dark cycle, lights on: 07:00, food and water ad libitum). All procedures involving animals and animal care were conducted in accordance with international laws and policies (Directive 2010/63/EU of the European parliament and of the council of 22 September 2010 on the protection of animals used for scientific purposes; Guide for the Care and Use of Laboratory Animals, U.S. National Research Council, 2011) and were approved by the Austrian Ministry of Science. All efforts were taken to minimize the number of animals used and their suffering.

### Lab-Inns2
Male 3-month-old 129S1/SvImJ (S1) mice (Charles River, Sulzfeld, Germany) were housed (four per cage) in a temperature- (22 ± 2°C) and humidity- (50–60%) controlled vivarium with food and water ad libitum under a 12 hr light/dark cycle. All mice were healthy and pathogen-free, with no obvious behavioral phenotypes. The Austrian Animal Experimentation Ethics Board (Bundesministerium für Wissenschaft Forschung und Wirtschaft, Kommission für Tierversuchsangelegenheiten) approved all experimental procedures.

### Lab-Wue1
All experiments were in accordance with the Guidelines set by the European Union and approved by our institutional Animal Care, the Utilization Committee and the Regierung von Unterfranken, Würzburg, Germany (License number: 55.2–2531.01-95/13). C57BL/6J wildtype mice were bred in the animal facility of the Institute of Clinical Neurobiology, University Hospital of Würzburg, Germany. Mice were housed in groups of three to five animals under standard laboratory conditions (12 hr/12 hr light/dark cycle, food and water ad libitum). All mice were healthy and pathogen-free, with no obvious behavioral phenotypes. Mice were quarterly tested according to the Harlan 51M profile (Harlan Laboratories, Netherlands). Yearly pathogen-screening was performed according to the Harlan 52M profile. All behavioral experiments were performed with male mice at an age of 8–12 weeks during the subjective day-phase of the animals and were randomly allocated to experimental groups.

## Zebrafish
### Lab-Wue2
Zebrafish (*Danio rerio*) embryos of the AB/AB strain (European Zebrafish Resource Center, Karlsruhe, Germany) were cultivated at 28°C with a 14/10 hr light/dark cycle in Danieau's solution containing 0.2 mM phenylthiocarbamide to prevent pigmentation. The embryos were staged according to *Kimmel et al., 1995*. To knock-down expression of *gad1b*, fertilized eggs were injected with 500 µM of a *gad1b* splice blocking morpholino targeting the exon 8/intron 8 boundary of *gad1b* (Ensembl, GRCz11). Morpholino sequence: 5'tttgtgatcagtttaccaggtgaga3' (Gene Tools). The efficiency of the morpholino was tested by reverse transcription PCR on DNase I treated total RNA collected from 24 hr post-fertilization and 5 days post-fertilization embryos. Sanger sequencing showed that the morpholino causes a partial inclusion of intron 8, which generates a premature stop codon.

## Mouse behavior
### Restraint stress and Pavlovian fear conditioning
### Lab-Mue
Animals were randomly assigned to four groups considering the following conditions; stress vs. control and early retrieval vs. late retrieval. Mice experienced restraint stress and a Pavlovian fear-

conditioning paradigm as described earlier *Chauveau et al., 2012*. In brief, on day one, animals of the stress group underwent restraint stress for 2 hr by using a perforated standard 50 ml falcon tube, allowing ventilation, but restricting movement. Animals of the control group remained in their homecages. On day 10, animals were adapted through two presentations of six $CS^-$ (2.5 kHz tone, 85 dB, stimulus duration 10 s, inter-stimulus interval 20 s; inter-trial interval 6 hr). On the next day, fear conditioning was performed in two sessions of three randomly presented $CS^+$ (10 kHz tone, 85 dB, stimulus duration 10 s, randomized inter-stimulus interval 10–30 s; inter-session interval 6 hr), each of which was co-terminated with an unconditioned stimulus (scrambled foot shock of 0.4 mA, duration 1 s). Animals of the early retrieval group underwent a retrieval phase on the same day (day 11), 1 hr after the last conditioning session, whereas animals of the late retrieval group underwent the retrieval phase on the next day (day 12), 24 hr after the conditioning session. For fear memory retrieval, mice were transferred to a new context. After an initial habituation phase of 2 min, mice were exposed to 4 $CS^-$ and 40 s later to 4 $CS^+$ presentations (stimulus duration 10 s, inter-stimulus interval 20 s) without receiving foot shocks. Afterwards, mice remained in this context for another 2 min before being returned to their homecages.

## Fear conditioning and extinction
### Lab-Inns1

Mice were single housed and stored in the experimental rooms in cages covered by filter tops with food and water ad libitum 3 days before behavioral testing. Fear acquisition and fear extinction were performed in a fear conditioning arena consisting of a transparent acrylic rodent conditioning chamber with a metal grid floor (Ugo Basile, Italy). Illumination was 80 lux and the chambers were cleaned with 70% ethanol. On acquisition day, following a habituation period of 120 s, mice were fear conditioned to the context by delivery of 5 foot-shocks (unconditioned stimulus, US, 0.5 mA for 2 s) with a random inter-trial interval of 70–100 s. After the test, mice remained in the test apparatus for an additional 120 s and were then returned to their homecage. On the next day, fear extinction training was performed. For this, mice were placed into the same arena as during acquisition and left undisturbed for 25 min. Freezing behavior was recorded and quantified by a pixel-based analysis software in one min bins (AnyMaze, Stoelting, USA). 90 min after the end of the extinction training, the mice were killed and the brains were processed for immunohistochemistry. Mice for homecage condition were kept in the experimental rooms for the same time period.

### Lab-Inns2

Cued fear conditioning, extinction and extinction retrieval was carried out as previously described *Whittle et al., 2016*. Context dependence of fear extinction memories was assessed using a fear renewal tests in a novel context *Murphy et al., 2017*. Fear conditioning and control of stimulus presentation occurred in a TSE operant system (TSE, Bad Homburg, Germany). Mice were conditioned in a 25 × 25 × 35 cm chamber with transparent walls and a metal-rod floor, cleaned with water, and illuminated to 300 lux ('context A'). The mice were allowed to acclimatize for 120 s before receiving three pairings of a 30 s, 75 dB 10 kHz sine tone conditioned stimulus (CS) and a 2 s scrambled-foot-shock unconditioned stimulus (US; 0.6 mA), with a 120 s inter-pairing interval. After the final pairing, mice were given a 120 s stimulus-free consolidation period before they were returned to the home-cage. Fear extinction training was performed in 'context B', a 25 × 25 × 35 cm cage with a solid gray floor and black walls, cleaned with a 100% ethanol and illuminated to 10 lux with a red lamp. After a 120 s acclimation period, the mice were subjected to 16x CS-alone trials, separated by 5 s inter-CS intervals. Extinction retrieval was conducted in 'context B' by repeating the conditions used in extinction training procedure but presenting only two CS trials. Fear renewal in a novel context was quantified 11 days following the extinction-retrieval test in a novel context ('context C'), a round plexiglas cylinder of 20 cm in diameter, and a height of 35 cm. The cylinder was covered on the outside with red diamond-printed white paper with an uneven pale ceramic tiled floor, illuminated to 5 lux with white light. After the mice were acclimated for 120 s, they were given two CS-alone trials, with a 5 s inter-CS interval. A trained observer blind to the animals' grouping measured freezing, defined as showing no visible movement except that required for respiration, as an index of fear *Gruene et al., 2015*. The observer manually scored freezing based on video recordings throughout the CS and determined the duration of freezing within the CS per total time of the CS in percent.

Freezing during all phases was averaged over two CS presentations and presented in eight trial blocks during extinction training and a one trail block each for extinction retrieval and fear renewal. Freezing during fear conditioning was quantified and presented as single CS. In the treatment group, mice which displayed freezing levels two times above than the standard deviation of the mean were classified as non-responders.

## Contextual fear conditioning
### Lab-Wue1
Male animals, initially kept as siblings in groups, were put to a new cage and kept in single-housing conditions with visual, olfactory and auditory contact in a ventilated cabinet (Scantainer, Scanbur). To habituate the mice to the male experimenter and the experimental rooms, mice were handled twice a day for at least 2 consecutive days prior to behavioral analysis. Mice were put to three different groups: (1) the homecage group, (2) the context control group that experienced the training context, but did not receive any electric foot shock, and (3) the context-conditioned group, which received electric foot shocks in the training context. Contextual fear (threat) conditioning was performed in a square conditioning arena with a metal grid floor (Multi conditioning setup, 256060 series, TSE, Bad Homburg, Germany). Before each experiment, the arena was cleaned with 70% ethanol. Mice were transported in their homecage to the experimental rooms and were put into the conditioning arena. After an initial habituation phase of 60 s, fear acquisition was induced by five electric foot shocks (unconditioned stimulus, US; 1 s, 0.7 mA) with an inter-stimulus interval of 60 s. After the foot shock presentation, mice remained in the training context for 30 s before being returned to their homecages in their housing cabinet. For fear memory retrieval, 24 hr after the training session, the mice were re-exposed to the conditioning arena for 360 s, without any US presentation. Mice were again put back to their homecage for 90 min, before mice were anesthetized and prepared for immunohistological analysis. Mouse behavior was videotaped. The MCS FCS-SQ MED software (TSE, Bad Homburg, Germany) was used to automatically track mice behavior and to quantify the freezing behavior during all sessions. Freezing was defined as a period of time of at least 2 s showing absence of visible movement except that required for respiration (*Fanselow, 1980*). The percentage time spent freezing was calculated by dividing the amount of time spent in the training chamber.

## Drug treatment
MS-275 (Entinostat, Selleck Chemicals, Vienna, Austria; 10 mg kg$^{-1}$ dissolved in saline +25% dimethylsulfoxide vehicle) was administered immediately (<1 min) following an extinction training session and L-DOPA (Sigma-Aldrich, Vienna, Austria; 20 mg kg$^{-1}$ dissolved in saline) was administered 1 hr before an extinction training session. All drugs were administered intraperitoneally in a volume of 10 ml kg$^{-1}$ body weight. Control animals received saline. Mice were randomly selected to be administered either vehicle or pharmacological compound (*Whittle et al., 2016*).

## Immunohistochemistry and microscopy
### Lab-Mue
Mice were anesthetized via inhalation anesthesia (isoflurane, 5% in O$_2$; CP Pharma, Germany) and perfused with phosphate-buffered saline (PBS) and then 4% paraformaldehyde (PFA; Roti-Histofix 4%, Carl Roth). Brains were isolated and post-fixed overnight in 4% PFA, treated with 30% sucrose/PBS solution for 48 hr, and then stored at 4°C until sectioning. Coronal sections (40 μm thick) were prepared on a freezing microtome (Leica, Wetzlar, Germany) and stored in PBS until use. Immunostaining was performed on free-floating sections. Sections were washed 3 × 10 min with PBS and then incubated in blocking solution (10% goat serum, 3% BSA, 0.3% Triton X100 in PBS) for 1 hr. After blocking, sections were treated at 4°C overnight with a primary antibody (rabbit anti-cFOS, 1:500, Santa Cruz Biotechnology, California, USA) diluted in blocking solution. On the next day, sections were washed 3 × 10 min with PBS and incubated for 1 hr at room temperature with the secondary antibody (goat anti-rabbit Alexa Fluor 488, 1:1000; Invitrogen, Germany) diluted in blocking solution. The incubation was followed by three 5 min washing steps in PBS. Sections were then mounted on SuperFrostPlus slides (Menzel, Braunschweig, Germany) and embedded with Vectashield Mounting Medium (Vector Laboratories, Burlingame, California) + 4',6-diamidino- 2-

phenylindole (DAPI). Fluorescence labeling was visualized and photographed using a laser-scanning confocal microscope (Nikon eC1 plus) with a 16x water objective at a step size of 1.5 µm, covering the whole section. Identical exposure settings were used for images that show the same region in the brains. The experimenter was blinded to the treatment conditions.

## Lab-Inns1

Ninety minutes after extinction training, mice were injected intraperitoneally with thiopental (150 mg/kg, i.p., Sandoz, Austria) for deep anesthesia. Transcardial perfusion, 3 min with PBS at room temperature followed by 10 min of 4% PFA at 4℃, was performed by a peristaltic pump at a flow rate of 9 ml/min (Ismatec, IPC, Cole-Parmer GmbH, Wertheim, Germany). Subsequently, brains were removed and postfixed in 4% PFA for 90 min at 4℃, cryoprotected for 48 hr in 20% sucrose at 4℃ and then snap frozen in isopentane (2-methylbutane, Merck GmbH, Austria) for 3 min at −60℃. Brains were transferred to pre-cooled open tubes and stored at −70℃ until further use. For immuno-histochemistry, coronal 40 µm sections were cut by a cryostat from rostral to caudal, collected in Tris-buffered saline (TBS) + 0.1% sodium azide. Sections from Bregma −1.22 mm (*Franklin and Paxinos, 2008*) were incubated for 30 min in TBS-Triton (0.4%), for 90 min in 10% normal goat/horse serum and overnight with the first primary antibody (diluted in 10% serum containing 0.1% sodium azide). Rabbit anti-cFOS (Millipore, PC-38, 1:20,000) and mouse anti-Parvalbumin (Sigma-Aldrich, P3088, 1:2500) were used as primary antibodies. After washing with TBS-buffer 3 × 5 min, secondary antibodies (goat anti-rabbit, Vector Laboratories inc, PI-1000, 1:1000 and biotinylated horse anti-mouse, Vector Laboratories inc, PK-4002, 1:200) were added to the sections for 150 min. Then, sections were incubated in the dark for 8 min in TSA-fluorescein (in-house, 1:100) staining solution (50 mM PBS and 0.02% $H_2O_2$). Sections were rinsed 3 × 5 min in TBS buffer and then incubated for 100 min in a solution of streptavidin Dylight 649 (Vector laboratories, SA5649, 1:100) in TBS buffer. Fluorescently stained sections were mounted on slides using gelatin and cover-slipped with glycerol-DABCO anti-fading mounting medium. Photomicrographs were taken on a fluorescent microscope (Zeiss Axio Imager M1) equipped with a halogen light source, respective filter sets and a monochrome camera (Hamamatsu ORCA ER C4742-80-12AG). Images of the basolateral amygdala (BLA) were taken with an EC Plan-Neofluar 10x/0.3 objective. All images were acquired using the same exposure time and software settings and the experimenter was blinded to the treatment conditions (homecage vs extinction).

## Lab-Inns2

Mice were killed 2 hr after the start of the fear renewal session using an overdose of sodium pentobarbital (200 mg/kg) and transcardially perfused with 40 ml of 0.9% saline followed by 40 ml of 4% paraformaldehyde in 0.1 M phosphate buffer, pH 7.4. Brains were then removed and post fixed at 4℃ for 2 hr in 4% paraformaldehyde in phosphate buffer. Brains were sectioned at the coronal plane with a thickness of 40 µm on a vibratome (VT1000S, Leica). Free-floating sections were incubated for 30 min in blocking solution using 1% BSA in 50 mM Tris buffer (pH 7.4) with 0.1% Triton-X100 and incubated overnight at 4℃ with a rabbit antibody against cFOS (1:1000; sc-52, Szabo-Scandic, Vienna, Austria). The sections were then washed (3 × 15 min in 1% BSA in Tris buffer containing 0.1% Triton-X100) and incubated for 2.5 hr with a secondary CY2-conjugated donkey anti rabbit IgG (1:500, #82371, Jackson ImmunoResearch). The sections were then washed (3 × 15 min in 50 mM Tris buffer), mounted on microscope slides and air-dried. Slides were embedded in ProLong Gold anti-fade reagent containing DAPI (P36935, Life Technologies). Immunofluorescence was assessed using a fluorescent microscope (Olympus BX51 microscope, Olympus XM10 video camera, CellSens Dimension 1.5 software, Olympus). Immunolabeled sections were visualized using a 20x oil-objective (UPlanSApo, Olympus) at 488 nm excitation.

## Lab-Wue1

To analyze anti-cFOS immunoreactivity after retrieval of a contextual memory, mice were anesthetized 90 min after the end of the retrieval session (C+). Mice that spent the same time in the conditioning arena without presentation of the US served as context controls (C-). Single-housed mice that were never exposed to the conditioning arena served as naïve learning controls (homecage; H). A rodent anesthesia setup (Harvard Apparatus) was used to quickly anesthetize the mice with the

volatile narcotic isoflurane (airflow 0.4 L/min, 4% isoflurane, Iso-Vet, Chanelle) for one minute. Then a mixture of ketamine (120 mg/kg; Ketavet, Pfizer) and xylazine (16 mg/kg; cp-Pharma, Xylavet, Burgdorf, Germany) was injected (12 μl/g bodyweight, intraperitoneal) to provide sedation and analgesia. Then anesthetized mice were transcardially perfused (gravity perfusion) with 0.4% heparin (Ratiopharm) in phosphate-buffered saline (PBS), followed by fixation with 4% paraformaldehyde in PBS. Brains were dissected and post-fixed in 4% paraformaldehyde for two hours at 4°C. The tissue was embedded in 6% agarose and coronal sections (40 μm) were cut using a vibratome (Leica VT1200). A total of 30 sections starting from Bregma −1.22 mm (*Franklin and Paxinos, 2008*) were considered as dorsal hippocampus. Immunofluorescent labeling was performed in 24-well plates with up to three sections per well under constant shaking. Slices were first incubated in 100 mM glycine, buffered at pH 7.4 with 2 M Tris-base for 1 hr at room temperature. Slices were transferred in blocking solution consisting of 10% normal horse serum, 0.3% Triton X100, 0.1% Tween 20 in PBS for 1 hr at room temperature. Primary antibodies were applied in blocking solution for 48 hr at 4°C. The following primary antibodies were used at indicated dilutions: mouse anti-Parvalbumin, SWANT, PV235, 1:5,000; guinea-pig anti-NeuN, SynapticSystems, 266004, 1:400; rabbit anti-cFOS, Synaptic-Systems, 226003, 1:10,000 (lot# 226003/7). Secondary antibodies were used for 1.5 hr at room temperature at a concentration of 0.5 μg / ml in blocking solution. The following antibodies were used: goat anti-mouse Alexa-488 conjugated (Life sciences, Thermo), donkey anti-rabbit Cy3 conjugated (Jackson ImmunoResearch), and donkey anti-guinea-pig Cy5 conjugated (Jackson ImmunoResearch). Sections were embedded in Aqua-Poly/Mount (Polysciences). Confocal image acquisition was performed with an Olympus IX81 microscope combined with an Olympus FV1000 confocal laser scanning microscope, a FVD10 SPD spectral detector and diode lasers of 473, 559, and 635 nm. Image acquisition was performed using an Olympus UPlan SAPO 20x/0.75 objective. Images with 1024 pixel to monitor 636 μm$^2$ were taken as 12 bit z-stacks with a step-size of 1.5 μm, covering the whole section. Images of *dentate gyrus* (DG), *Cornu ammonis 1* (CA1) and CA3 were taken in each hemisphere of three sections of the dorsal hippocampus to achieve a maximum of six images (n) per region for each animal (N). During image acquisition, the experimenter was blinded to the treatment condition (C+ versus C- versus H).

## Lab-Wue2

For immunohistochemistry of whole-mount specimens, the embryos were fixed at 30 hr post-fertilization in 4% PFA at 4°C over night. The specimens were subsequently washed 3 × 10 min in PBS with 0.1% Tween-20 (PBST) and then once for 5 min in 150 mM Tris-HCl buffer (pH 9.0). The solution was exchanged for fresh Tris-HCl buffer, and the embryos were incubated for 15 min at 70°C, cooled down to room temperature and then washed 2 × 5 min in PBST. To further increase permeability, the embryos were first rinsed quickly two times in ice-cold dH$_2$O and then incubated with pre-cooled acetone for 20 min at −20°C. The acetone was quickly washed off with dH$_2$O, and then with PBS containing 0.8% Triton X100 (PBSTX) for 2 × 5 min. The specimens were incubated at room temperature for 1 hr in blocking buffer (PBSTX with 10% normal sheep serum and 2% bovine serum albumin) and subsequently with the primary antibody (rabbit-anti-GABA, Sigma-Aldrich A2052, diluted 1:400 in blocking buffer) at 4°C for 3 days with gentle shaking. After extensive washes in PBSTX, the embryos were incubated with the secondary antibody (goat-anti-rabbit AlexaFluor488, Invitrogen, Thermo Fisher Scientific, diluted 1:1000 in blocking buffer) at 4°C for 2 days with gentle shaking. Finally, the embryos were washed extensively in PBST, transferred and stored in 80% glycerol in PBST at 4°C until imaging. Before microscopy, the yolk was removed and the embryos were mounted in 80% glycerol to be imaged from the dorsal side. Confocal image acquisition was performed using a Nikon eclipse C1 laser scanning microscope with a Plan Apo VC 20x/0.75 DIC N2 objective and a Coherent Saphire 488 nm laser. All specimens were imaged using NIS Elements software (Nikon) with the same acquisition settings. Images with 2048x2048 pixels were taken as 12 bit z-stacks with a step-size of 2.5 μm, covering the whole region, including the dorsal-ventral dimension, of the hindbrain that contains GABA immunoreactive cells.

## Image processing and manual analysis

### Lab-Mue

Images were adjusted in brightness and contrast using ImageJ. One expert from Lab-Mue manually segmented the paraventricular thalamus (PVT) and quantified cFOS-positive cells in the PVT for bio-image analysis. For the training and validation of DL models, cFOS-positive ROIs in five additional images were manually segmented by four experts from *Lab-Wue1* (expert 1 and experts 3–5) using ImageJ. All experts were blinded to another and the treatment condition.

### Lab-Inns1

Number of cFOS-positive neurons was obtained from two basolateral amygdalae (BLA) per animal of five homecage mice and five mice subjected to contextual fear extinction. PV staining was used to identify the localization and extension of the BLA and the borders were manually drawn by a neuroscientist of Lab-Inns1 using the free shape tool of the Improvision Openlab software (PerkinElmer). Boundaries were projected to the respective cFOS-immunoreactive image and cFOS-positive nuclei were counted manually inside that area by the expert of Lab-Inns1. For the training and validation of DL models, cFOS-positive ROIs in five additional images were manually segmented by four experts from *Lab-Wue1* (expert 1 and experts 3–5) using ImageJ. All experts were blinded to another and the treatment condition.

### Lab-Inns2

The anatomical localization of cells within the infralimbic cortex was aided by using illustrations in a stereotaxic atlas (*Franklin and Paxinos, 2008*), published anatomical studies (*Van De Werd et al., 2010*) and former studies in S1 mice (*Fitzgerald et al., 2014*; *Whittle et al., 2010*). All analyses were done in a comparable area under similar optical and light conditions. For manual analysis, an expert of Lab-Inns2 viewed the digitized images on a computer screen using CellSens Dimension 1.5 software (Olympus Corporation, Tokyo, Japan) and evaluated cFOS-positive nuclei within the infralimbic cortex, the brain region of the interest. For the training and validation of DL models, cFOS-positive ROIs in five additional images were manually segmented by four experts from *Lab-Wue1* (expert 1 and experts 3–5) using ImageJ. All experts were blinded to another and the treatment condition.

### Lab-Wue1

For image preprocessing, 12-bit gray-scale confocal image z-stacks were projected (maximum intensity) and converted to 8-bit. Five expert neuroscientists from *Lab-Wue1* manually segmented cFOS-positive nuclei and Parvalbumin-positive somata as regions of interest (ROIs) in a total of 45 images (36 training and nine test images). The NeuN immunoreactive granule cell layer of the dentate gyrus and the pyramidal cell layer in CA1 and CA3 were annotated as NeuN-positive ROIs. The NeuN-positive areas used for the quantifications of cFOS-positive cells were identical for all analyses and segmented manually by one human expert. All experts were blinded to another and the treatment condition.

### Lab-Wue2

The confocal z-stacks with 12-bit gray-scale images were imported into ImageJ (*Schneider et al., 2012*). An expert of *Lab-Wue2* manually counted GABA-positive somata in every 4th section of each confocal z-stack, covering the entire hindbrain region housing GABAergic neurons ($N_{untreated\ controls} = 5$, $N_{morphants} = 4$). For the training and validation of DL models, GABA-positive ROIs in five additional images were manually segmented by three experts from *Lab-Wue2* (experts 6–8) using ImageJ. All experts were blinded to another and the treatment condition.

## Ground truth estimation

In absence of an objective ground truth, we derived a probabilistic estimate of the ground truth by running the expectation-maximization algorithm for simultaneous truth and performance level estimation (STAPLE, *Warfield et al., 2004*). The STAPLE algorithm iteratively estimates the ground truth

segmentation (est. GT) based on the expert segmentation maps. During each algorithm iteration two steps are performed:

**Estimation step:** The ground truth segmentation's conditional probability is estimated based on the expert decisions and previous performance parameter estimates.

**Maximization step:** The performance parameters (sensitivity and specificity) for each expert segmentation are estimated by maximizing the conditional expectation.

Iterations are repeated until convergence is reached. We implemented the algorithm using the simplified interface to the Insight Toolkit (SimpleITK 1.2.4, *Lowekamp et al., 2013*).

## Evaluation metrics

All evaluation metrics were calculated using Python (version 3.7.3), SciPy (version 1.4.1), and scikit-image (version 0.16.2). We provide the source code in *Jupyter Notebooks* (see 7.13 Data and software availability).

### Segmentation and detection

Following *Caicedo et al., 2019* we based our evaluation on identifying segmentation and detection similarities on object-level (ROI-level). In a segmentation mask, we define an object as a set of pixels that were horizontally, vertically, and diagonally connected (8-connectivity). We only considered ROIs at a biologically justifiable size, depending on the data set characteristics. We approximated the minimum size based on the smallest area that was annotated by a human expert (*Lab-Mue*: 30px, *Lab-Inns1*: 16px, *Lab-Inns2*: 60px, *Lab-Wue1*: 30px, *Lab-Wue2*: 112px).

To compare the segmentation similarity between a source and a target segmentation mask, we first computed the intersection-over-union (IoU) score for all pairs of objects. The IoU, also known as Jaccard similarity, of two sets of pixels $a = \{1, ..., A\}$ and $b = \{1, ..., B\}$ is defined as the size of the intersection divided by the size of the union:

$$M_{\mathrm{IoU}}(a,b) := \frac{|a \cap b|}{|a \cup b|}$$

Second, we used the pairwise IoUs to match the objects of each mask. We solved the assignment problem by maximizing the sum of IoUs by means of the Hungarian method (*Kuhn, 1955*). This ensures an optimal matching of objects in the case of ambiguity, that is, overlap of one source object with one or more targets object. We reported the segmentation similarity of two segmentation masks by calculating the arithmetic mean of $M_{\mathrm{IoU}}$ over all matching objects:

$$\bar{M}_{\mathrm{IoU}} = \frac{1}{N} \sum_{i=1}^{N} M_{\mathrm{IoU}}^{i}$$

where $i \in \{0, ..., N\}$ is an assigned match and $N$ denotes number of matching objects. By this definition, the *Mean IoU* only serves as a measure for the segmentation similarity of matching objects and neglects objects that do not overlap at all.

To address this issue, we additionally calculated measures to account for the detection similarity. Therefore, we define a pair of objects with an IoU is above a threshold $t$ as correctly detected (true positive - *TP*). Objects that match with an IoU at or below $t$ or have no match at all are considered to be false negative (*FN*) for the source mask and false positive (*FP*) for the target mask. This allows us to calculate the Precision $M_{\mathrm{Precision}}$, Recall $M_{\mathrm{Recall}}$, and F1 score $M_{\mathrm{F1score}}$ as the harmonic mean of $M_{\mathrm{Precision}}$ and $M_{\mathrm{Recall}}$:

$$M_{\mathrm{Precision}}(t) := \frac{TP(t)}{TP(t) + FP(t)}$$

$$M_{\mathrm{Recall}}(t) := \frac{TP(t)}{TP(t) + FN(t)}$$

$$M_{\text{F1score}}(t) := 2 \cdot \frac{M_{\text{Precision}}(t) \cdot M_{\text{Recall}}(t)}{M_{\text{Precision}}(t) + M_{\text{Recall}}(t)}$$

with $t \in [0,1]$ as a fixed IoU threshold. If not indicated differently, we used $t = 0.5$ in our calculations.

### Inter-rater reliability

To quantify the reliability of agreement between different annotators we calculated Fleiss' $\kappa$ (*Fleiss and Cohen, 1973*). In contrast to the previously introduced metrics, Fleiss' $\kappa$ accounts for the agreement that would be expected by chance. For a collection of segmentation masks of the same image, each object (ROI) $i \in \{1, ..., N\}$ is assigned to a class $j \in \{0, ..., K\}$. Here, $N$ denotes the total number of unique objects (ROIs) and $K$ the number of categories ($K = 1$ for binary segmentation). Then, $n_{ij}$ represents the number of annotators who assigned object $i$ to class $j$. We again leveraged the IoU metric to match the ROIs from different segmentation masks above a given threshold $t \in [0,1]$. Following *Fleiss and Cohen, 1973*, we define the proportion of all assignments for each class:

$$p_j(t) := \frac{1}{Nd} \sum_{i=1}^{N} n_{ij}(t)$$

where $d$ denotes the count of the annotators. We define the extent to which the annotators agree on the i-th object as

$$P_i(t) := \frac{1}{d(d-1)} \sum_{j=1}^{K} n_{ij}(t)\big(n_{ij}(t) - 1\big)$$

Subsequently, we define the mean of the $P_i(t)$ as

$$\bar{P}(t) := \frac{1}{N} \sum_{i=1}^{N} P_i(t)$$

and

$$\bar{P}_e(t) := \sum_{j=1}^{K} p_j(t)^2$$

Finally, Fleiss' $\kappa$ at a given threshold $t$ is defined as

$$\kappa(t) := \frac{\bar{P}(t) - \bar{P}_e(t)}{1 - \bar{P}_e(t)}$$

where $1 - \bar{P}_e(t)$ denotes the degree of agreement attainable above chance and $\bar{P}(t) - \bar{P}_e(t)$ the actually achieved agreement in excess of chance. To allow a better estimate of the chance we randomly added region proposals of class $j = 0$ (background). If not indicated differently, we use $t = 0.5$ in our calculations.

## Deep learning approach

The deep learning pipeline was implemented in Python (version 3.7.3), Tensorflow (version 1.14.0), Keras (version 2.2.4), scikit-image (version 0.16.2), and scikit-learn (version 0.21.2). We provide the source code in *Jupyter Notebooks* (see 7.13 Data and software availability).

### Network architecture

We instantiated all DL models with a U-Net architecture (*Ronneberger et al., 2015*), a fully convolutional neural network for semantic segmentation. The key principle of a U-Net is that one computational path stays at the original scale, preserving the spatial information for the output, while the other computational path learns the specific features necessary for classification by applying convolutional filters and thus condensing information (*Ronneberger et al., 2015*). We adopted the model hyperparameters (e.g. hidden layers, activation functions, weight initialization) from *Falk et al., 2019*

as these are extensively tested and evaluated on different biomedical data sets. The layers of the U-Net architecture are logically grouped into an encoder and a decoder (*Figure 1—figure supplement 1*). Following *Falk et al., 2019* the VGG-like encoder consists of five convolutional modules. Each module comprises two convolution layers with no padding, each followed by a leaky ReLU with a leakage factor of 0.1 and a max-pooling operation with a stride of two. The last module, however, does not contain the max-pooling layer and constitutes the origin of the decoder. The decoder consists of four (up-) convolutional modules. Each of these modules comprises of a transposed convolution layer (also called up- or deconvolution), a concatenate layer for the corresponding cropped encoder feature map, and two convolution layers. Again, each layer is followed by a leaky ReLU with a leakage factor of 0.1. The final layer consists of a $1 \times 1$ convolution with a softmax activation function. The resulting (pseudo-) probabilities allow a comparison to the target segmentation mask using cross-entropy on pixel level. Unless indicated differently, we used a kernel size of $3 \times 3$. To allow faster convergence during training, we included batch normalization layers (*Ioffe and Szegedy, 2015*) after all (up-) convolutions below the first level. By this, an unnormalized path from the input features to the output is remaining to account for absolute input values, for example, the brightness of fluorescent labels.

## Weighted soft-max cross-entropy loss

Fluorescent microscopy images typically exhibit more background than fluorescent features of interest. To control the impacts of the resulting class imbalance, we implemented a pixel-weighted soft-max cross-entropy loss. Thus, we compute the loss from the raw score (logits) of the last $1 \times 1$ convolution without applying the softmax. As proposed by *Falk et al., 2019* we define the weighted cross entropy loss for an input image $I$ as

$$L_{\text{wce}}(I) := -\sum_{x \in \Omega} w(x) \log \frac{\exp(\hat{y}_{y(x)}(x))}{\sum_{k=0}^{K} \exp(\hat{y}_k(x))}$$

where $x$ is a pixel in image domain $\Omega$, $w : \Omega \to \mathbb{R}_{\geq 0}$ the pixel-wise weight map, $y : \Omega \to \{0, ..., K\}$ the target segmentation mask, $\hat{y}_k : \Omega \to \mathbb{R}$ the predicted score for class $k \in \{0, ..., K\}$, and $K$ the number of classes ($K = 1$ for binary classification). Consequently, $\hat{y}_{y(x)}(x)$ is the predicted score for the target class $y(x)$ at position $x$.

Similar to *Falk et al., 2019* we compose the weight map $w$ from two different weight maps $w_{bal}$ and $w_{sep}$. The former allows to mitigate the class imbalances by decreasing the weight of background pixels by the factor $v_{\text{bal}} \in [0, 1]$. We add a smoothly decreasing Gaussian function at the edges of the foreground objects accordingly and define

$$w_{bal}(x) := \begin{cases} 1 & y(x) > 0 \\ v_{\text{bal}} + (1 - v_{bal}) \exp\left(-\frac{d_1^2(x)}{2\sigma_{\text{bal}}^2}\right) & y(x) = 0 \end{cases}$$

where $d_1(x)$ denotes the distance to the closest foreground object and $\sigma_{\text{bal}}$ the standard deviation of the Gaussian function.

By definition, semantic segmentation performs a pixel-wise classification and is unaware of different object instances (ROIs). Following *Falk et al., 2019*, we the force learning of the different instances by increasing the weight of the separating ridges. We estimate the width of a ridge by adding $d_1$ (distance to nearest ROI) and $d_2$ (distance to second nearest ROI) at each pixel. We define

$$w_{\text{sep}}(x) := \exp\left(-\frac{(d_1(x) + d_2(x))^2}{2\sigma_{\text{sep}}^2}\right)$$

where $\sigma_{\text{sep}}$ defines the standard deviation of the decreasing Gaussian function. The final weight map is given by $w := w_{\text{bal}} + \lambda w_{\text{sep}}$ where $\lambda \in \mathbb{R}_{\geq 0}$ allows to control the focus on instance separation. We used the following parameter set in our experiments: $\lambda = 50$, $v_{bal} = 0.1$, $\sigma_{\text{bal}} = 10$ and $\sigma_{\text{sep}} = 6$.

## Tile sampling and augmentation

Given limited training data availability, we leveraged effective data augmentation techniques for biomedical images as proposed by *Falk et al., 2019*. These comprise transformations and elastic

deformations by means of a random deformation field. To become invariant to the input sizes (image shapes), we leveraged the overlap tile strategy introduced by *Ronneberger et al., 2015*. Thus, images of any size can be processed. Both data augmentation and overlap tile strategy were adopted from a Tensorflow implementation of *Falk et al., 2019*. We used an input tile shape of 540 $\times$ 540 $\times$ 1 (height x width x channels) and a corresponding output tile shape of 356 $\times$ 356 $\times$ 1 for all our experiments.

## Training, evaluation, and model selection

We trained, evaluated and selected all deep learning models for our different strategies – *expert models*, *consensus models*, *consenus ensembles* – following the same steps:

1. Determining an appropriate learning rate using the *learning rate finder* (*Smith, 2018*)
2. Splitting the data into train and validation set (random stratified sampling)
3. Training the model on the train set according to the *fit-one-cycle* policy of *Smith, 2018*
4. Selecting the model with the highest $M_{\mathrm{F1score}}$ median on the validation set (post-hoc evaluation).

We used the annotations from individual experts to train the *expert models* and the consensus annotations (est. GT) for the *consensus models* and *consensus ensembles*. The post-hoc evaluation on the validation set was performed using the saved model weights (checkpoints) from each epoch. For the similarity analysis, we converted the model output (pixel-wise softmax score) to a segmentation mask by assigning each pixel to the class with the highest softmax score. For the *consensus ensemble* approach, we repeated the steps above according to the principle of a *k-fold cross-validation*. We ensembled the resulting *k* models by averaging the softmax predictions.

Our initial experimental results have indicated that an adequately trained DL-model performs on par with a human expert. However, insufficient training data may impair the model performance. As there were only five annotated training images for the external laboratories (*Lab-Mue*, *Lab-Inns1*, *Lab-Inns2*, and *Lab-Wue2*), we additionally defined a model selection criterion to establish trust in our consensus ensemble approaches: A selected consensus model must at least match the performance of the 'worst' human expert for each validation image (measured as the $M_{\mathrm{F1score}}$ to the estimated ground truth). This selection criterion serves as a lower bound for individual model performance. All consensus models trained for *Lab-Wue1* met this criterion. For the other laboratories, we have included the model selection results in *Figure 5—source data 3*. In those cases where the criterion discarded models, the issue was typically due to a validation image being very different from the training data for a given train-validation split. This issue was often resolved when pretrained model weights were used. For the *frozen* approach (see 7.10.5 Transfer learning) the models never met the selection criterion. Yet, we decided to retain these models to facilitate a comparison among the different approaches. We also indicated that these models and ensembles should be considered with caution and did not use them for further biological analyses.

We trained all models on a NVIDIA GeForce GTX 1080 TI with 11 GB GDDR5X RAM using the Adam optimizer (*Kingma and Ba, 2014*) and a mini-batch size of four. If not indicated differently, the initial weights were drawn from a truncated normal distribution (*He et al., 2015*). We chose the appropriate maximum learning rates according to the learning rate finder (step two). For *Lab-Wue1* we selected a maximum learning rate of 4e-4 and a minimum learning rate of 4e-5 over a training cycle length of 972 iterations within $k = 4$ validation splits. For *Lab-Mue, Lab-Inns1, Lab-Inns2* and *Lab-Wue2* we chose a maximum learning rate of 1e-4 and a minimum learning rate of 1e-5 over a training cycle length of 972 iterations within $k = 5$ validation splits.

## 7.10.5 Transfer learning

To implement transfer learning we adapted the training procedure from above. For the *fine-tuning* approach, we initialized the weights from the *consensus models* of *Lab-Wue1* and performed all steps for model training, evaluation and selection. For the *frozen* approach, we also initialized the weights from the *consensus models* of *Lab-Wue1* but skipped steps two (finding a learning rate) and three (model training). Hence, we did not adjust the model weights to the new training data. Hardware and training hyperparameters remained unchanged.

## Quantification of fluorescent features

Fluorescent features were analyzed on base of the binary segmentation masks derived from the output of DL models or model ensembles, or counted manually by lab-specific experts. In order to compare the number of fluorescent features across images, we normalized in each image the number of annotated fluorescent features to the area of the analyzed region (e.g. the number of cFOS-positive features per NeuN-positive area for *Lab-Wue1*). For one set of experiment, we pooled this data for each condition (e.g. H, C- and C+ for *Lab-Wue1*) and the analyzed brain region (e.g. whole DG, infrapyramidal DG, suprapyramidal DG, CA3, or CA1 for *Lab-Wue1*). To compare different sets of experiments with each other, we normalized all relative fluorescent feature counts to the mean value of the respective control group (e.g. H for *Lab-Wue1*).

The mean signal intensity for each image was calculated as the mean signal intensity of all ROIs annotated within the analyzed NeuN-positive region (only performed for *Lab-Wue1*). Subsequent pooling steps were identical as described above for the count of fluorescent features.

## Statistical analysis

All statistical analyses were performed using Python (version 3.7.3), SciPy (version 1.4.1), and Pingouin (version 0.3.4). We provide all source datasets and source codes in *Jupyter Notebooks* (see 7.13 Data and software availability). In box plots, the area of the box represents the interquartile range (IQR, 1st to 3rd quartile) and whiskers extend to the maximal or minimal values, but no longer than $1.5 \times$ IQR.

### 7.12.1 Statistical analysis of fluorescent feature quantifications

All DL-based quantifications of fluorescent features were tested for significant outliers (Grubb's test). If an image was detected as significant outlier in several DL-based quantification results, it was visually inspected by an expert and excluded from the analysis if abnormalities (e.g. clusters of fluorescent particles or folding of the tissue) were detected. Throughout all bioimage analyses, N represents the number of investigated animals and n the number of analyzed images. Normality (Shapiro-Wilk) and homogeneity of variance (Levene's) were tested for all DL-based quantification results. For comparison of multiple quantifications of the same image dataset, non-parametric statistical tests were applied to all bioimage analyses. This ensured comparability of the results. To compare two experimental conditions (*Lab-Mue, Lab-Inns1*, and *Lab-Wue2*), Mann-Whitney-U tests were used. In case of three experimental conditions (*Lab-Wue1* and *Lab-Inns2*), Kruskal-Wallis-ANOVA followed by Mann-Whitney-U tests with Bonferroni correction for multiple comparisons was applied.

### 7.12.2 Effect size calculation

Effect sizes ($\eta^2$) were calculated for each pairwise comparison. First, the Z-statistic (*Z*) was calculated from the U-statistic (*U*) of the Mann-Whitney-U test as:

$$Z = \frac{U - \frac{n_1 \cdot n_2}{2}}{\sqrt{\frac{n_1 \cdot n_2 \cdot (n_1 + n_2 + 1)}{12}}}$$

where $n_1$ and $n_2$ are the numbers of analyzed images of the two compared groups, group 1 and group 2, respectively. Following *Rosenthal and DiMatteo, 2002*, $\eta^2$ was calculated as:

$$\eta^2 = \frac{Z^2}{n_1 + n_2}$$

Furthermore, the three critical values of $\eta^2$ that mark the borders between the four significance levels (e.g. for p = 0.05, p = 0.01, and p = 0.001 for a pairwise comparison without Bonferroni correction for multiple comparisons) were calculated from the chi-square distribution.

### All other statistical analyses

Data was tested for normal distribution (Shapiro-Wilk) and homoscedasticity (Levene's) and parametric or non-parametric tests were used accordingly, as reported in the figure legends (parametric: one-way ANOVA, followed by T-tests (or Welchs T-test for unequal sample sized) with Bonferroni

correction for multiple comparisons; non-parametric: Kruskal-Wallis ANOVA, followed by Mann-Whitney tests with Bonferroni correction for multiple comparisons).

## Data and software availability

### Data

We provide the full bioimage datasets of *Lab-Wue1* and *Lab-Mue*, including microscopy images, segmentation masks of all DL models and ensembles, and annotations of analyzed regions of interest. For all five bioimage datasets, we provide the quantification results of the bioimage analyses for all models and ensembles as source data. Likewise, the results of the behavioral analysis of *Lab-Wue1* are available as source data. As indicated in the respective figure legends, we also provide all statistical data in full detail as source data files. Furthermore, we provide all training and test datasets that were created in the course of this study. These include all microscopy images with the corresponding manual expert annotations and estimated ground-truth annotations. As part of our proposed pipeline, we share one trained and validated consensus ensembles for each bioimage dataset within our open-source model library. All data and code can be accessed at our Dryad repository (www.doi.org/10.5061/dryad.4b8gtht9d). The source code is also available in our GitHub repository (www.github.com/matjesg/bioimage_analysis; *Segebarth, 2020*; copy archived at swh:1: rev:eafeb5f8e1312ab29416144df0212761ddf4cfc4).

### Software

We provide all source code within python modules and Jupyter Notebooks in our Dryad (www.doi. org/10.5061/dryad.4b8gtht9d) and in our GitHub repository (www.github.com/matjesg/bioimage_ analysis). This includes the code for the bioimage analyses, all statistical analyses, and our proposed pipeline to create, use, and share consensus ensembles for fluorescent feature annotations.

## Acknowledgements

We thank all members of our collaborative research center (SFB-TRR58) for all the fruitful discussions and the continuous support over the course of this project. We thank Toni Greif for critically reviewing the mathematical content. We thank Thorsten Falk for providing us with the code for data pre-processing and augmentation. We thank Friederike Griebel for her valuable advice on the design of our central figures.

## Additional information

### Funding

| Funder | Grant reference number | Author |
| --- | --- | --- |
| Deutsche Forschungsgemeinschaft | ID 44541416 - TRR58 A10 | Robert Blum |
| Deutsche Forschungsgemeinschaft | ID 44541416 - TRR58 A03 | Hans-Christian Pape |
| Deutsche Forschungsgemeinschaft | ID 44541416 - TRR58 B08 | Maren D Lange |
| Graduate School of Life Sciences Wuerzburg | fellowship | Rohini Gupta Manju Sasi |
| Austrian Science Fund | P29952 & P25851 | Ramon O Tasan |
| Austrian Science Fund | I 3875 | Nicolas Singewald |
| Austrian Science Fund | DKW-1206 | Nicolas Singewald |
| Austrian Science Fund | SFB F4410 | Nicolas Singewald |
| Interdisziplinaeres Zentrum fuer Klinische Zusammenarbeit Wuerzburg | N-320 | Christina Lillesaar |
| Deutsche Forschungsge- | ID 424778381 A02 | Robert Blum |

meinschaft

The funders had no role in study design, data collection and interpretation, or the decision to submit the work for publication.

### Author contributions
Dennis Segebarth, Matthias Griebel, Conceptualization, Data curation, Software, Formal analysis, Validation, Investigation, Visualization, Methodology, Writing - original draft, Project administration, Writing - review and editing; Nikolai Stein, Conceptualization, Validation, Visualization, Methodology, Writing - original draft, Project administration; Cora R von Collenberg, Dominik Fiedler, Lucas B Comeras, Anupam Sah, Victoria Schoeffler, Teresa Lüffe, Data curation, Validation, Investigation, Methodology, Writing - review and editing; Corinna Martin, Data curation, Validation, Writing - review and editing; Alexander Dürr, Software, Writing - review and editing; Rohini Gupta, Manju Sasi, Data curation, Funding acquisition, Writing - review and editing; Christina Lillesaar, Resources, Data curation, Funding acquisition, Validation, Methodology, Project administration, Writing - review and editing; Maren D Lange, Conceptualization, Resources, Data curation, Funding acquisition, Validation, Investigation, Methodology, Project administration, Writing - review and editing; Ramon O Tasan, Nicolas Singewald, Resources, Funding acquisition, Validation, Methodology, Project administration, Writing - review and editing; Hans-Christian Pape, Conceptualization, Resources, Supervision, Funding acquisition, Validation, Methodology, Writing - original draft, Project administration; Christoph M Flath, Conceptualization, Resources, Supervision, Funding acquisition, Validation, Methodology, Writing - original draft, Project administration, Writing - review and editing; Robert Blum, Conceptualization, Resources, Supervision, Funding acquisition, Validation, Investigation, Visualization, Methodology, Writing - original draft, Project administration, Writing - review and editing

### Author ORCIDs
Dennis Segebarth (iD) https://orcid.org/0000-0002-3806-9324
Matthias Griebel (iD) https://orcid.org/0000-0003-1959-0242
Lucas B Comeras (iD) https://orcid.org/0000-0003-2445-3605
Anupam Sah (iD) https://orcid.org/0000-0001-8298-6501
Christina Lillesaar (iD) http://orcid.org/0000-0002-5166-2851
Nicolas Singewald (iD) http://orcid.org/0000-0002-0166-3370
Hans-Christian Pape (iD) http://orcid.org/0000-0001-6874-8224
Christoph M Flath (iD) https://orcid.org/0000-0002-1761-9833
Robert Blum (iD) https://orcid.org/0000-0002-5270-3854

### Ethics
Animal experimentation: All effort was taken to minimize the number of animals used and their suffering. Lab-Wue1: All experiments with C57BL/6J wildtype mice were in accordance with the Guidelines set by the European Union and approved by our institutional Animal Care, the Utilization Committee and the Regierung von Unterfranken, Würzburg, Germany (License number: 55.2-2531.01-95/13). C57BL/6J wildtype mice were bred in the animal facility of the Institute of Clinical Neurobiology, University Hospital of Würzburg, Germany. Lab Mue: All animal experiments with male C57Bl/6J mice (Charles River, Sulzfeld, Germany) were carried out in accordance with European regulations on animal experimentation and protocols were approved by the local authorities (Landesamt für Natur, Umwelt und Verbraucherschutz Nordrhein-Westfalen). Lab-Inns1: Experiments were performed in adult, male C57Bl/6NCrl mice (Charles River, Sulzfeld, Germany). They were bred in the Department of Pharmacology at the Medical University Innsbruck, Austria. All procedures involving animals and animal care were conducted in accordance with international laws and policies (Directive 2010/63/EU of the European parliament and of the council of 22 September 2010 on the protection of animals used for scientific purposes; Guide for the Care and Use of Laboratory Animals, U.S. National Research Council, 2011) and were approved by the Austrian Ministry of Science. Lab-Inns2: Male 129S1/SvImJ (S1) mice (Charles River, Sulzfeld, Germany) were used for experimentation. The Austrian Animal Experimentation Ethics Board (Bundesministerium für Wissenschaft Forschung und Wirtschaft, Kommission für Tierversuchsangelegenheiten) approved all experimental procedures.

Decision letter and Author response
Decision letter https://doi.org/10.7554/eLife.59780.sa1
Author response https://doi.org/10.7554/eLife.59780.sa2

## Additional files

### Supplementary files
• Transparent reporting form

### Data availability

The official repository of our study "On the objectivity, reliability, and validity of deep learning enabled bioimage analyses" can be found at Dryad (https://doi.org/10.5061/dryad.4b8gtht9d). In addition, we also provide all code in our GitHub repository (https://github.com/matjesg/bioimage_analysis). A copy is archived at https://archive.softwareheritage.org/swh:1:rev:eafeb5f8e1312ab29416144df0212761ddf4cfc4/.

The following dataset was generated:

| Author(s) | Year | Dataset title | Dataset URL | Database and Identifier |
|---|---|---|---|---|
| Segebarth D, Griebel M, Stein N, Martin C, Fiedler D, Comeras LB, Sah A, Schoeffler V, Lüffe T, Dürr A, Gupta R, Sasi M, Lillesaar C, Lange MD, Tasan RO, Singewald N, Pape HC, Flath CM, Blum R, von Collenberg CR | 2020 | Data from: On the objectivity, reliability, and validity of deep learning enabled bioimage analyses | https://doi.org/10.5061/dryad.4b8gtht9d | Dryad Digital Repository, 10.5061/dryad.4b8gtht9d |

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
