## [Decision Letter]

**Acceptance summary:**

There are three things about this contribution that make it certain to impact the field in a positive fashion. The clear presentation style, the balanced discussion and that careful benchmarking should establish a more objective style of evaluation. Given the speed with which this field is moving and the need for better standards, this should become a well-cited paper.

**Decision letter after peer review:**

[Editors’ note: the authors submitted for reconsideration following the decision after peer review. What follows is the decision letter after the first round of review.]

Thank you for submitting your work entitled "DeepFLaSH, a deep learning pipeline for segmentation of fluorescent labels in microscopy images" for consideration by *eLife*. Your article has been reviewed by three peer reviewers, and the evaluation has been overseen by a Reviewing Editor and a Senior Editor.

Our decision has been reached after consultation between the reviewers. Based on these discussions and the individual reviews below, we regret to inform you that your work will not be considered further for publication in *eLife*.

The reviewers and I appreciate the strengths of DeepFLaSH, and the challenges of the important problem you are attacking. We all feel that research employing fluorescence imaging will be dramatically enhanced as optimized machine learning techniques are refined that are tolerant of the particular challenges you outline. However, the current manuscript falls short of documenting the advance represented by DeepFLaSH.

We feel that any study reporting an advance must fully embrace two key demonstrations:

1) A complete and balanced comparison of the speed, ease-of-use and accuracy of the new approach (in this case, DeepFLaSH) with a full spectrum of the other tools currently available. The reviewers believe that the comparisons performed in this manuscript are not sufficient in depth or breadth. Comparisons to techniques that would not be employed by a knowledgeable user because they are known to fail (eg. a simple thresholding) only serves to weaken the confidence of the reader in the nature of the advance.

2) A use of the new approach (in this case, DeepFLaSH) to analyze a biological system and extract a publishable advance.

A manuscript expanded in this fashion, to perform a relevant bank of tests and to present a novel biological insight, would be high-impact, and would justify publication in *eLife* or other first-tier journals.

Reviewer #1:

The paper presents DeepFlaSH, a tool to apply U-Net networks to microscopy images. Although the presented experimental results are interesting, I think that the manuscript has several significant issues that have to be addressed before it can be considered for publication in *eLife*. Most importantly, the focus of the manuscript is unclear, and the presented tool does not seem to be significantly different from existing tools. More detailed comments are given below:

– It is unclear what the focus of the manuscript is. The title and manuscript type (Tools and Resources) suggest that the focus is on the introduction of the DeepFlaSH tool, but the manuscript itself is much more focused on the specific application of studying contextual fear conditioning. There is little description of the tool, how it can be used, and how it relates to existing tools and resources. Instead, a detailed description of the experiments and their results is given. Presenting these experimental results could be an interesting paper, since the results show that U-Net networks can be successfully applied for fluorescence labeling in real-world experiments, but in that case the title, manuscript type, and focus of the paper should be adjusted to reflect this.

– Related, since the current focus of the paper is the DeepFlaSH tool, I would expect a relatively mature software tool that is significantly different from existing tools/resources, for example by providing an integrated tool for manual labeling, network training, and application of trained networks to new images. In the current form, however, the DeepFlaSH tool is not much more than a single example Python script that calls a slightly modified U-Net network implemented with Keras. This is not significantly different from code packages that are released with many machine learning papers, and, in fact, there are already many existing packages that provide U-Net-like network implementations (in Keras and other frameworks) with accompanying example scripts. Therefore, I don't really know whether the presented tool is novel enough for a paper in *eLife*, and, in any case, I don't think the authors can claim that the tool is a “unique deep learning pipeline”.

– The use of Google Colab to offer a cloud-based solution that users can run the code in is useful, but also introduces several potential issues. First, the service could be changed, stopped, and/or monetized at any time by Google, which could limit the DeepFlaSH tool itself. Second, resources of Google Colab servers have to be shared between users, and at a given moment, required resources such as a GPU could be either unavailable or only available in a limited way (several reports can be found online about this). Finally, to be able to use DeepFlaSH in Google Colab, the user has to upload their data to a Google server, which users might not feel comfortable with and/or are unable to do because of privacy concerns. Because of these issues, I am not sure whether the tool can be presented as a “cloud-based virtual notebook with free access to high computing power”, unless a cloud-based solution is made available that is not as dependent on a specific company.

– Related, the authors state that the tool can be run on a local machine, but the code itself doesn't really support this at the moment (although it is technically possible). No installation instructions are given, no list of dependencies are given, and the code itself currently depends on Google Colab functionality and is not available as an easily installable and callable Python module.

Reviewer #2:

In this manuscript Segebarth et al. describe the application of a variant of the U-Net deep learning network to segmentation of cells in histological brain slices. They show segmentation in 2D on two markers, one localizing with nuclei and one showing the activity of a particular signal in the soma. They use standard tricks of data augmentation by image transformation to increase the pool of training data, although it is not documented quantitatively how necessary this is.

Overall, there is nothing severely wrong with this work, but there is also nothing new or exciting. The authors mention that the segmentation pipeline called DeepFlaSH is accessible through a cloud-based notebook, which offers simple operation. I did not test this. It may be possible that DeepFlaSH is particularly simple to use, and thus a brief application note may be justified. Although, such note would have to better acknowledge the numerous existing software tools addressing particle / nucleus detection, which are integrated in software platforms that are widely accessed by the community and probably are, I quote the authors, “hassle-free” in their use as well. I am not sure whether *eLife* is in the business of documenting random software packages. I am sure that DeepFlasSH serves the authors lab very well, and the tool could be advertised in the methods sections of papers describing the application in a particular project, even as an accessible open-source. It will then be picked up by the community working on similar data. But, DeepFlaSH does not hold up for a standalone paper on histological image segmentation. As a regular author and reviewer I have always seen *eLife* as a journal publishing scientific discoveries or innovative methods that have the promise to lead to such discoveries. I am afraid, the present manuscript does not meet that standard.

From an image analysis perspective, I would actually challenge the authors with the claim that any filter, potentially for the soma segmentation a multi-scale filter, will do an equal or better segmentation job than DeepFlaSH. The authors anecdotally compare the Deep Learning result to a trivial thresholding method. This is inappropriate. No one with a bit of understanding of signal processing expects a simple threshold to work on the data presented in Figure 1. But, there is an extended literature on particle detection that relies on linear or non-linear amplification of bona fide particle signals followed by, in the more advanced approaches, statistics-based foreground-background segregation. The advantage of such an approach is that the segmentation is unsupervised and controlled by an interpretable confidence level for signal-to-background separation. In my view the presented segmentation problem does not require Deep Learning and I would suspect that conventional particle detectors will actually do better, as they explicitly include priors of the punctuate nature of the signal.

From the perspective of the Deep Learning (DL) field, even the DL subfield focused on applications in microscopy, there is nothing new to be found in this paper. The authors make a point about integration of training sets from multiple trainers to increase robustness. In my view this is a standard measure in supervised machine learning. Also, there is no systematic performance analysis (e.g. what are the break points with respect to abundance or SNR of the training data) or interpretation of what that network learns and how variation, e.g. simplification, of the network affects the segmentation.

Reviewer #3:

This paper presents a method for the automatic segmentation of fluorescence microscopy images from brain specimens. The method is trained by and validated against ground truth image segmentations of five experts. A cloud-based implementation of the segmentation pipeline is available for public use.

The method ("DeepFLaSH") is based on the U-Net deep convolutional neural network architecture, but with two alterations that purportedly reduce the amount of training data required: batch normalization and depthwise separable convolution. The network performs on par with experts, after having been trained on just 36 1024x1024 images. The method appears to be solid and certainly of utility to others.

The paper falls short in one key area. This is in explicitly comparing DeepFLaSH's performance to a traditional U-Net. For readers to benefit fully from this work, they should better understand its method's improvement over techniques they would otherwise use. I would suggest:

1) The authors train a U-Net on the same data to demonstrate the performance difference.

2) Stating the number of network parameters that are determined through training, both for DeepFLaSH and the corresponding U-Net.

3) Show how the network performance depends on the size of the training set, from just a few images up to the full 36. Has cross-validation performance saturated?

[Editors’ note: further revisions were suggested prior to acceptance, as described below.]

Thank you for submitting your article "On the objectivity, reliability, and validity of deep learning enabled bioimage analyses" for consideration by *eLife*. Your article has been reviewed by three peer reviewers, and the evaluation has been overseen by a Reviewing Editor and Marianne Bronner as the Senior Editor. The following individuals involved in review of your submission have agreed to reveal their identity: Dan Ruderman (Reviewer #1); Gaudenz Danuser (Reviewer #3).

The reviewers have discussed the reviews with one another and the Reviewing Editor has drafted this decision to help you prepare a revised submission. We all feel that this manuscript will be ready for acceptance after some minor revisions, not requiring re-review.

I agree with all three reviewers that this manuscript is a significant step forward. Reviewer 3 summarized the manuscript and its strengths nicely: "This is a very high quality evaluation of procedures that should be put in place to set up objective, reliable, and valid Deep Learning models for fluorescence image segmentation. I commend the authors for teaming up multiple labs in order to build consensus “ground truth” models. They have done an impressive amount of work in documenting the performance of the various strategies for generating training data sets. The paper is written very well. Despite the dryness of the subject and the at times dense lines of argumentation, one can get through the material in a reasonable amount of time, and then go deep where necessary. This paper can become the reference for every investigator starting a quantitative imaging study with deep learning."

The reviewers and I would like to offer some suggested changes to the presentation, without requiring the addition of more experimental work or analyses:

1) Figures and Tables.

A) Both reviewer 2 and I found Figure 1 a bit confusing and I found at least three different ways that it could be misunderstood. The problem is that the workflow icons in the individual vertical boxes appear to lie up with the horizontal bars, which I do not believe is your intent. Please revise Figure 1 so that your intent is more clear. I suggest that changing the scale and spacing of the icons might solve it, but putting the workflow icons below the horizontal bars would make the figure only slightly larger and make your intent much more clear.

B) Please use similar scales and formats for figures that readers are likely to compare, and might misunderstand. For example Figure 2—figure supplement 3 and Figure 2—figure supplement 4 are slightly different sizes, and use different scales (what the different shades of gray represent), so I found myself puzzled. Figure 5—figure supplement 3 and Figure 5—figure supplement 4 use different scales as well.

C) Table 1, and elsewhere in the text needs attention to the number of significant figures that are displayed. Table 1 shows p-values to 5 or 4 significant figures for non-significant differences. It shows p-values to 3 significant figures for some of the significant differences. If the authors reflect on this, I believe they will agree that there is no way that the data set is sufficiently large for this number of significant figures. Furthermore, the large number of figures distract the reader from the message I believe the authors are intending to convey.

2) Language and definitions.

A) The reviewers point out that this very strong paper is not made stronger by claims that seem beyond beyond the data. These include the claim that the DL approach surpasses the experts.

B) There are terms that should be clarified or defined; in most of these cases, it might be wise to find language that is less likely to be misunderstood. These include: "invalid", "bad", "biased", "irreproducible", "appear to alter the results", "DL models are proven to…" etc. Note that all of these should be easy to resolve, but could confuse, anger and/or frustrate a reader, which will not optimize the acceptance of the important lessons of this work.

Reviewer #1:

This work is high quality, employs real-world data sets from multiple sources, and addresses questions of broad interest. I recommend publication in *eLife*.

Reviewer #2:

In general, I really enjoyed reading this manuscript, and I think the new focus resulted in a much clearer paper compared with the previous version. As a researcher in machine learning myself, the manuscript includes several findings that I will use in my future research, especially the comparison between the various training approaches. As such, I think that the paper will be a valuable contribution to the field, and I recommend publication after my comments below (and those of other reviewers) have been addressed. Note that my expertise is in developing machine learning algorithms, and I don't have enough background in the specific biology application to feel comfortable with commenting on the validity of the biology.

1) My main concern with respect to the method is the exclusion of what the authors call “invalid” methods in the ensemble approaches. By construction, this exclusion biases the results towards better metrics. Therefore, it is not entirely clear whether the better performance of the ensemble approaches is caused by the ensemble itself, or rather simply by the fact that multiple networks were trained and “bad” networks were thrown away. This question can be answered in several ways: for example, it would already be informative to know how many networks were rejected in this way (if it is a small number compared with the total number of networks, it is not likely the improvement is due to the rejection). Even better would be to also apply the same rejection strategy to the expert model approach and see whether that improves these by the same amount (even though you would have access to the estimated GT in pure expert model applications).

2) My second main concern is with regards to practical application of the consensus ensemble approach. In many applications, it is very time consuming to acquire manually annotated images due to the required expertise for manual annotation and the time it takes to annotate. Therefore, getting enough manual annotations to obtain accurate consensus models might be prohibitively difficult in practice. A solution for this might be to use multiple experts, but have each expert annotate a different set of images. This would drastically reduce the required manual annotation time compared with obtaining a full consensus model, but you would still have information from multiple experts, which might improve results. For the current manuscript, it would be very informative to include this approach in the results. One way of doing this is to use the data that the authors already have, assign each of the 36 training images to one of the 5 experts, and then during training only use the manual annotation of the assigned expert for that image.

3) To me Figure 1 is quite hard to read. I do get what the authors mean, but the fact that the icons in the “vertical blocks” (e.g. “data annotation” and “automated annotation”) align with the rows makes it seem that each icon in the block actually belongs to a certain row. A solution would be to rearrange the icons inside each block somewhat (e.g. by making them smaller) so that they don't line up anymore with the rows.

4) It would be interesting to investigate what the accuracy is of an approach in which only the GT of a single expert is used, but multiple networks are trained in an ensemble. In other words, a combination between the expert models and consensus ensembles, but without using the estimated GT. This would indicate whether the improvement of the consensus ensembles is not purely due to the ensemble itself. I don't expect that this will actually achieve very accurate results, so probably a short paragraph or an added sentence or two will be enough to describe these additional results in the paper.

Reviewer #3:

This is a very high quality evaluation of procedures that should be put in place to set up objective, reliable, and valid Deep Learning models for fluorescence image segmentation. I commend the authors for teaming up multiple labs in order to build consensus “ground truth” models. They have done an impressive amount of work in documenting the performance of the various strategies for generating training data sets. The paper is written very well. Despite the dryness of the subject and the at times dense lines of argumentation, one can get through the material in a reasonable amount of time, and then go deep where necessary. This paper can become the reference for every investigator starting a quantitative imaging study with deep learning. I really have nothing substantial to comment.

---

## [Author Response]

[Editors’ note: the authors resubmitted a revised version of the paper for consideration. What follows is the authors’ response to the first round of review.]

The reviewers and I appreciate the strengths of DeepFLaSH, and the challenges of the important problem you are attacking. We all feel that research employing fluorescence imaging will be dramatically enhanced as optimized machine learning techniques are refined that are tolerant of the particular challenges you outline. However, the current manuscript falls short of documenting the advance represented by DeepFLaSH.We feel that any study reporting an advance must fully embrace two key demonstrations:1) A complete and balanced comparison of the speed, ease-of-use and accuracy of the new approach (in this case, DeepFLaSH) with a full spectrum of the other tools currently available. The reviewers believe that the comparisons performed in this manuscript are not sufficient in depth or breadth. Comparisons to techniques that would not be employed by a knowledgeable user because they are known to fail (eg. a simple thresholding) only serves to weaken the confidence of the reader in the nature of the advance.2) A use of the new approach (in this case, DeepFLaSH) to analyze a biological system and extract a publishable advance.A manuscript expanded in this fashion, to perform a relevant bank of tests and to present a novel biological insight, would be high-impact, and would justify publication in eLife or other first-tier journals.

The thorough reviews have made it very clear to us that we did a subpar job of conveying the main intention behind the manuscript.

Our primary goal was not to establish a new method in the sense of a Deep Learning (DL) algorithm, but rather to demonstrate that a light-weight, end-to-end integration of DL methods can reliably and reproducibly verify the presence of biological effects in laboratory data – both on the inter-individual level (heterogeneous coding in lab) as well as on the inter-lab level (a network trained in lab A can with minimal re-training be used to analyze data from lab B).

Our work ultimately addresses a central concern put forward by Falk et al. (2019, Nature Methods: U-Net: deep learning for cell counting, detection, and morphometry): “U-Net learns from the provided examples. If the examples are not representative of the actual task, or if the manual annotation in these examples is low quality and inconsistent, U-Net will either fail to train or will reproduce inconsistent annotations on new data.” We suggest the following corollary of this statement: Local instantiations of deep learning models (e.g., training in a certain lab) can at most speed up the local analysis pipeline while retaining the intrinsic bias of human coders. Yet, a common integrated workflow instantiated by a light-weight tool (we called this DeepFlaSH) can additionally ensure objectivity, reliability, reproducibility and transparency through shared neural network weights.

To confirm this idea experimentally, we created typical image raw data and we used the biological model of cFOS changes after behavioral training of mice. There is no ground truth in these images showing fluorescent labels of cFOS; there is no ultimate parameter allowing signal definition. One cannot decide, whether “manual annotation in these examples is low quality and inconsistent” or not. Our experimental strategy gave us a second parameter; the behavior of the mice. Easy said: the analysis of the mouse behavior gave us a coincident parameter in order to decide whether the U-Net failed to create consistent or inconsistent data. Our main result is that the speed and flexibility of a light-weight DL workflow (such as DeepFlaSH) can be used for a higher reproducibility, reliability, objectivity and transparency in image analysis.

Over the last year, we moved forward and fully overhauled the entire study. We followed our initial idea, namely to investigate how DL can contribute to more objectivity and reproducibility of bioimage analyses. Considering these developments, we think that there is no benefit of associating this study with the prior submission.

[Editors’ note: what follows is the authors’ response to the second round of review.]

The reviewers and I would like to offer some suggested changes to the presentation, without requiring the addition of more experimental work or analyses:1) Figures and Tables.A) Both reviewer 2 and I found Figure 1 a bit confusing and I found at least three different ways that it could be misunderstood. The problem is that the workflow icons in the individual vertical boxes appear to lie up with the horizontal bars, which I do not believe is your intent. Please revise Figure 1 so that your intent is more clear. I suggest that changing the scale and spacing of the icons might solve it, but putting the workflow icons below the horizontal bars would make the figure only slightly larger and make you intent much more clear.

We appreciate this input. Having spent a lot of time on Figure 1 in the initial submission process we were so deep into it that we did not realize that the ordering of icons, process steps and approaches may be confusing. Looking at the figure again after 3 months, the issues you raised become directly evident. We modified the figure along your suggestions and feel that it is much easier to understand now.

B) Please use similar scales and formats for figures that readers are likely to compare, and might misunderstand. For example Figure 2—figure supplement 3 and Figure 2—figure supplement 4 are slightly different sizes, and use different scales (what the different shades of gray represent), so I found myself puzzled. Figure 5—figure supplement 3 and Figure 5—figure supplement 4 use different scales as well.

In our initial version we tried to exploit the whole greyscale space to visually distinguish the heatmap entries, using different scales for each metric (f1 score/iou). We agree that the resulting scaling differences lead to confusion when comparing across figures. We revised the figures and aligned the scales for Figure 2—figure supplement 3/ Figure 2—figure supplement 4 and Figure 5—figure supplement 3/ Figure 5—figure supplement 4.

C) Table 1, and elsewhere in the text needs attention to the number of significant figures that are displayed. Table 1 shows p-values to 5 or 4 significant figures for non-significant differences. It shows p-values to 3 significant figures for some of the significant differences. If the authors reflect on this, I believe they will agree that there is no way that the data set is sufficiently large for this number of significant figures. Furthermore, the large number of figures distract the reader from the message I believe the authors are intending to convey.

We fully agree that the limited size of these data sets also restricts the validity of the calculated p-values to less post decimal positions. We therefore reduced the number of presented post decimal positions for the calculated p-values in Table 1 to 3 and denote p-values that are smaller than 0.001 as < 0.001.

2) Language and definitions.A) The reviewers point out that this very strong paper is not made stronger by claims that seem beyond beyond the data. These include the claim that the DL approach surpasses the experts.B) There are terms that should be clarified or defined; in most of these cases, it might be wise to find language that is less likely to be misunderstood. These include: "invalid", "bad", "biased", "irreproducible", "appear to alter the results", "DL models are proven to…" etc. Note that all of these should be easy to resolve, but could confuse, anger and/or frustrate a reader, which will not optimize the acceptance of the important lessons of this work.

We appreciate these pointers to our usage of ambiguous terms. In the revised document we carefully rephrased the corresponding passages. We replaced the corresponding terms or replaced the sentences with more careful statements. Overstatements without solid statistical foundation such as “constantly outperform/surpass” were deleted.

Reviewer #1:This work is high quality, employs real-world data sets from multiple sources, and addresses questions of broad interest. I recommend publication in eLife.Reviewer #2:In general, I really enjoyed reading this manuscript, and I think the new focus resulted in a much clearer paper compared with the previous version. As a researcher in machine learning myself, the manuscript includes several findings that I will use in my future research, especially the comparison between the various training approaches. As such, I think that the paper will be a valuable contribution to the field, and I recommend publication after my comments below (and those of other reviewers) have been addressed. Note that my expertise is in developing machine learning algorithms, and I don't have enough background in the specific biology application to feel comfortable with commenting on the validity of the biology.1) My main concern with respect to the method is the exclusion of what the authors call “invalid” methods in the ensemble approaches. By construction, this exclusion biases the results towards better metrics. Therefore, it is not entirely clear whether the better performance of the ensemble approaches is caused by the ensemble itself, or rather simply by the fact that multiple networks were trained and “bad” networks were thrown away. This question can be answered in several ways: for example, it would already be informative to know how many networks were rejected in this way (if it is a small number compared with the total number of networks, it is not likely the improvement is due to the rejection). Even better would be to also apply the same rejection strategy to the expert model approach and see whether that improves these by the same amount (even though you would have access to the estimated GT in pure expert model applications).

Thank you for this valuable feedback! We agree that our initial manuscript did not sufficiently justify the model selection process and the impacts on model performance. Following your suggestion, we have indicated the number of discarded models in the corresponding figure legends (Figures 5, Figure 5—figure supplement 1, Figure 5—figure supplement 2) and included the model selection results in the corresponding figure source data. Moreover, we have revised the description of the model selection process (as stated in our response to the Editor).

2) My second main concern is with regards to practical application of the consensus ensemble approach. In many applications, it is very time consuming to acquire manually annotated images due to the required expertise for manual annotation and the time it takes to annotate. Therefore, getting enough manual annotations to obtain accurate consensus models might be prohibitively difficult in practice. A solution for this might be to use multiple experts, but have each expert annotate a different set of images. This would drastically reduce the required manual annotation time compared with obtaining a full consensus model, but you would still have information from multiple experts, which might improve results. For the current manuscript, it would be very informative to include this approach in the results. One way of doing this is to use the data that the authors already have, assign each of the 36 training images to one of the 5 experts, and then during training only use the manual annotation of the assigned expert for that image.

Indeed, the acquisition of annotated training data was one of the greatest challenges for our study, and it will most likely continue to be the bottleneck for future DL based studies. For this study, we avoided splitting the training images because our aim was to control the DL strategies exactly on the same images to get information about the variability between experts and expert models. During our experiments, we have already tried different approaches for model training, e.g., we trained models using all expert segmentation instead of the estimated ground truth. This approach led to an unstable training behavior (heavy oscillations of the loss function from iteration to iteration). We would argue that your suggested approach could lead to the same training instability (of course, depending on the data and the differences between expert annotations). However, we agree that there is still potential to reduce the annotation effort that should be evaluated in future studies.

3) To me Figure 1 is quite hard to read. I do get what the authors mean, but the fact that the icons in the “vertical blocks” (e.g. “data annotation” and “automated annotation”) align with the rows makes it seem that each icon in the block actually belongs to a certain row. A solution would be to rearrange the icons inside each block somewhat (e.g. by making them smaller) so that they don't line up anymore with the rows.

We modified the figure and feel that it is much easier to understand now.

4) It would be interesting to investigate what the accuracy is of an approach in which only the GT of a single expert is used, but multiple network are trained in an ensemble. In other words, a combination between the expert models and consensus ensembles, but without using the estimated GT. This would indicate whether the improvement of the consensus ensembles is not purely due to the ensemble itself. I don't expect that this will actually achieve very accurate results, so probably a short paragraph or an added sentence or two will be enough to describe these additional results in the paper.

As stated above, we trained models using all expert segmentation instead of the estimated ground truth in our initial experiments. This has led to an unstable training behavior. We think this was caused by the considerable differences in expert annotations. This approach, however, might work with less ambiguous data.